# NHE6 depletion corrects ApoE4-mediated synaptic impairments and reduces amyloid plaque load

**Theresa Pohlkamp**[1,2]*†, **Xunde Xian**[1,2,3]†, **Connie H Wong**[1,2]†,
**Murat S Durakoglugil**[1,2], **Gordon Chandler Werthmann**[1,2], **Takaomi C Saido**[4],
**Bret M Evers**[2], **Charles L White III**[5], **Jade Connor**[1,2], **Robert E Hammer**[6],
**Joachim Herz**[1,2,7,8]*

[1]Department of Molecular Genetics, University of Texas Southwestern Medical Center, Dallas, United States; [2]Center for Translational Neurodegeneration Research, Dallas, United States; [3]Institute of Cardiovascular Sciences and Key Laboratory of Molecular Cardiovascular Sciences, Ministry of Education, Peking University, Beijing, China; [4]Laboratory for Proteolytic Neuroscience, Riken Center for Brain Science, Wako, Japan; [5]Pathology, University of Texas Southwestern Medical Center, Dallas, United States; [6]Department of Biochemistry, University of Texas Southwestern Medical Center, Dallas, United States; [7]Department of Neuroscience, University of Texas Southwestern Medical Center, Dallas, United States; [8]Department of Neurology and Neurotherapeutics, University of Texas Southwestern Medical Center, Dallas, United States

*For correspondence:
ThePohlkamp@gmail.com (TP);
joachim.herz@utsouthwestern.
edu (JH)

†These authors contributed
equally to this work

Competing interest: See page
28

Reviewing Editor: Jeannie Chin,
Baylor College of Medicine,
United States

**Abstract** Apolipoprotein E4 (ApoE4) is the most important and prevalent risk factor for late-onset Alzheimer's disease (AD). The isoelectric point of ApoE4 matches the pH of the early endo-some (EE), causing its delayed dissociation from ApoE receptors and hence impaired endolysosomal trafficking, disruption of synaptic homeostasis, and reduced amyloid clearance. We have shown that enhancing endosomal acidification by inhibiting the EE-specific sodium-hydrogen exchanger 6 (NHE6) restores vesicular trafficking and normalizes synaptic homeostasis. Remarkably and unex-pectedly, loss of NHE6 (encoded by the gene *Slc9a6*) in mice effectively suppressed amyloid depo-sition even in the absence of ApoE4, suggesting that accelerated acidification of EEs caused by the absence of NHE6 occludes the effect of ApoE on amyloid plaque formation. NHE6 suppression or inhibition may thus be a universal, ApoE-independent approach to prevent amyloid buildup in the brain. These findings suggest a novel therapeutic approach for the prevention of AD by which partial NHE6 inhibition reverses the ApoE4-induced endolysosomal trafficking defect and reduces plaque load.

## Introduction

ApoE is the principal lipid transport protein in the brain. Three different ApoE isoforms are common in humans: ApoE2 (ε2), ApoE3 (ε3), and ApoE4 (ε4). Each ApoE4 allele reduces the age of Alzhei-mer's disease (AD) onset by approximately 3–5 years compared to ApoE3 homozygotes, which comprise ~80 % of the human population (*Roses, 1994*; *Sando et al., 2008*). By contrast and by comparison, ApoE2 is protective against AD (*Corder et al., 1994*; *Panza et al., 2000*; *West et al., 1994*). ApoE is an arginine-rich protein and a major component of very-low-density lipoproteins (*Shore and Shore, 1973*). The number of positively charged arginine residues differs between the three human isoforms due to two single nucleotide polymorphisms in the ApoE gene. The most

common isoform, ApoE3, has a charge neutral cysteine at amino acid position 112 and an arginine at position 158. The second most common isoform, ApoE4, has two arginines, while the less frequent ApoE2 has two cysteines at these respective positions. The positively charged arginines raise the net charge and thus the isoelectric point (IEP) of the protein (*Eto et al., 1985*; *Warnick et al., 1979*). The IEP of ApoE2 is the lowest (5.9), ApoE3 has an intermediate IEP of 6.1, and the IEP of ApoE4 is ~6.4 (*Ordovas et al., 1987*).

For cargo delivery, ApoE binds to lipoprotein receptors and undergoes endocytosis and recycling. Endocytic subcompartments become progressively more acidic, and the pH of these compartments is regulated by the opposing functions of vesicular ATP-dependent proton pumps (vATPase) and proton leakage channels (Na$^+$/H$^+$ exchangers, NHEs). Early endocytic vesicles are slightly acidic (pH, ~6.4), which facilitates ligand/receptor dissociation. Lysosomes are highly acidic (pH 4–5), which is required for the digestion of endocytosed biomolecules (*Figure 1A*; *Casey et al., 2010*; *Naslavsky and Caplan, 2018*). For maturation of the early endosomes (EEs) and entry into the next sorting stage, ligand/receptor dissociation is required. The pH-dependent release of ApoE from its receptor in the EE is important for endosomal maturation and cargo delivery (*Yamamoto et al., 2008*) for the ability of endosomal content to rapidly recycle to the cell surface (*Heeren et al., 1999*; *Nixon, 2017*). The early endosomal pH, which triggers ligand-receptor dissociation, closely matches the IEP of ApoE4 (*Xian et al., 2018*). Loss of net surface charge at the IEP is accompanied by reduced solubility in an aqueous environment, leading to impaired dissociation of ApoE4 from its receptors (*Xian et al., 2018*) and aided by a greater propensity of ApoE4 to form a molten globule configuration under acidic conditions (*Morrow et al., 2002*). Dysregulation of endolysosomal trafficking by ApoE4 causes an age-dependent increase in EE number and size (*Nuriel et al., 2017*).

Based on these observations, we have proposed a model in which destabilization of ApoE4 in the acidic EE environment, combined with a greater propensity for self-association, results in delayed detachment from its receptors (*Figure 1B*). Subsequent endosomal swelling through K$^+$, Na$^+$, and H$_2$O influx further impairs cargo delivery, receptor recycling, and ligand re-secretion. Importantly, in neurons, ApoE and its receptor Apoer2 travel together with glutamate receptors through the endosomal recycling pathway (*Chen et al., 2010*; *Xian et al., 2018*). Rapid endocytosis and subsequent recycling of synaptic receptors is triggered by the synaptic homeostatic modulator and Apoer2 ligand Reelin (*Hiesberger et al., 1999*; *Trommsdorff et al., 1999*). We previously showed that ApoE4, in contrast to ApoE3 and ApoE2, prevents Reelin-mediated glutamate receptor insertion at the synapse, a state we refer to as ApoE4-mediated Reelin resistance (*Chen et al., 2010*; *Durakoglugil et al., 2009*; *Lane-Donovan and Herz, 2017*; *Lane-Donovan et al., 2014*; *Xian et al., 2018*). Reduction of endosomal pH and increasing the differential to the ApoE4 IEP abolishes this effect in vitro (*Xian et al., 2018*).

The pH of EE compartments is controlled by the vATPase-dependent proton pump and proton leakage through NHE6 (*Nakamura et al., 2005*; *Basu et al., 1976*; *Davis et al., 1987*; *Rudenko et al., 2002*). NHE6 is encoded by the gene *Slc9a6*. We showed that EE acidification by pharmaceutical pan-NHE inhibition or selective NHE6/*Slc9a6* knockdown in neurons prevents the ApoE4-caused trafficking delay of ApoE and glutamate receptors (*Xian et al., 2018*). NHE6 deficiency in humans causes neurodevelopmental defects, which result in Christianson syndrome, an X-linked genetic disorder characterized by cognitive dysfunction, autism, ataxia, and epilepsy. However, some *Slc9a6* mutant variants causing Christianson syndrome in humans do not significantly alter the ion exchange properties of NHE6 (*Ilie et al., 2020*; *Ilie et al., 2019*) suggesting that Christianson syndrome could be caused by loss of NHE6 scaffolding functions and not by loss of endosomal pH regulation. To investigate whether NHE6 depletion can reverse ApoE4 pathology in vivo, we generated a conditional *Slc9a6* knockout (KO) mouse line (*Slc9a6$^{fl}$;CAG-Cre$^{ERT2}$*) to avoid complications caused by neurodevelopmental defects by temporally and spatially controlling NHE6 ablation. We show that genetic NHE6 ablation attenuates both, the ApoE4-induced Reelin resistance and impaired synaptic plasticity in ApoE4 targeted replacement (*Apoe$^{APOE4}$*) mice using hippocampal field recordings.

The pathological hallmarks of AD are extracellular aggregates of the amyloid β (Aβ) peptide and intracellular tangles of hyperphosphorylated tau protein. Processing of the transmembrane amyloid precursor protein (APP) at the β- and γ-sites leads to Aβ production. Aβ forms neurotoxic oligomers and accumulates in plaques. The β-site amyloid precursor protein cleaving enzyme 1 (BACE1) cleaves APP in its extracellular juxtamembrane domain to create a membrane-anchored C-terminal fragment

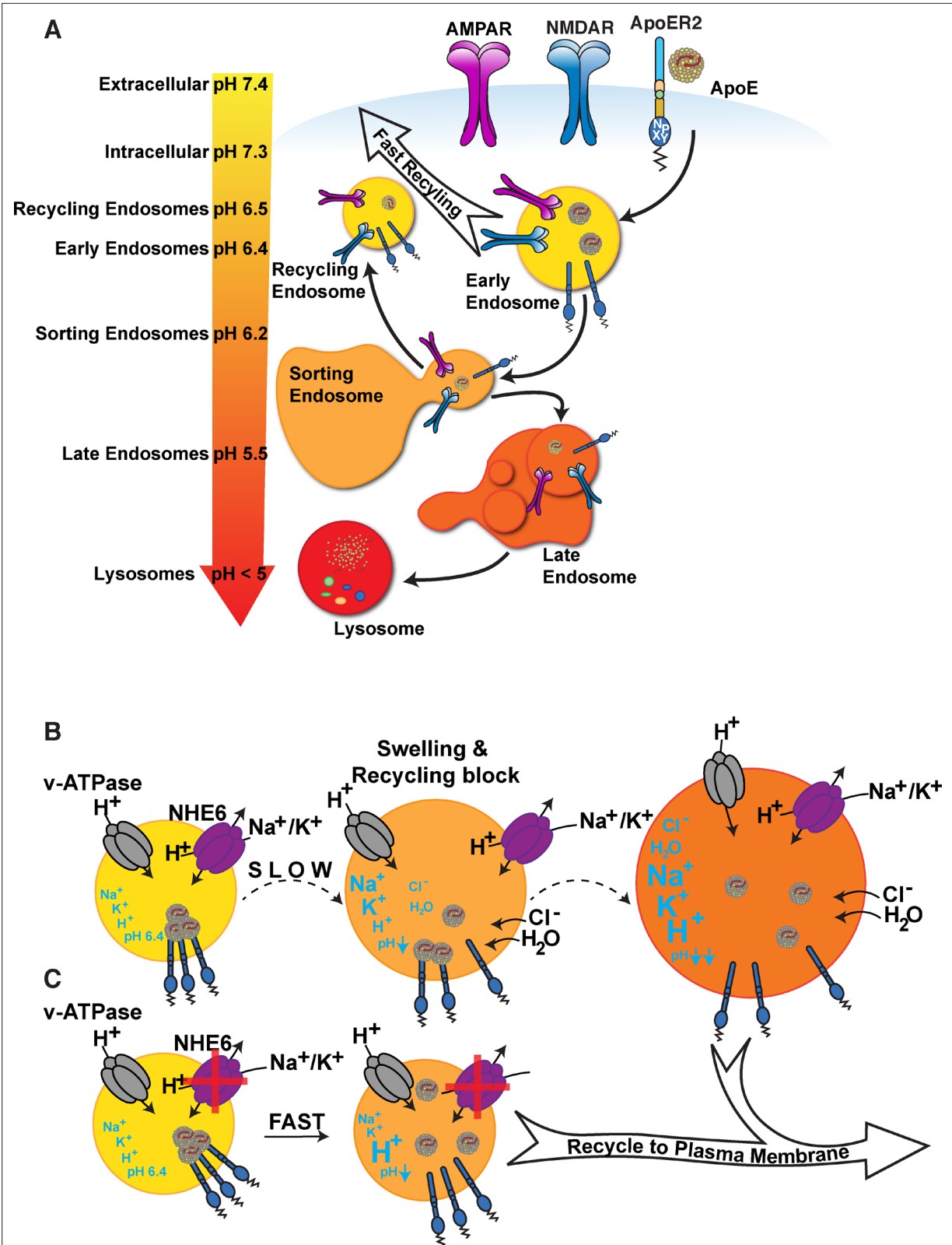

**Figure 1.** ApoE4 induces endolysosomal trafficking delay. (**A**) pH regulation within the endolysosomal pathway. Upon receptor binding, ApoE is endocytosed along with glutamate receptors (AMPA and NMDA receptors). Cargo that has entered the early endosome (EE) can undergo recycling through a fast direct route without further acidification (fast recycling) or through slower sorting pathways that require further acidification (*Casey et al., 2010*; *Naslavsky and Caplan, 2018*). While lipid components are targeted to the lysosome, the majority of receptors, as well as ApoE, remain in

*Figure 1 continued on next page*

*Figure 1 continued*

endosomal compartments at the cellular periphery where they rapidly move back to the surface (**Heeren et al., 1999**). The increasingly acidic luminal pH is illustrated as a color gradient and depicted on the left. (**B**) In the presence of ApoE4, early endosomal trafficking and fast recycling are delayed. At the pH of the EE, ApoE4 is near its isoelectric point where solubility is reduced (**Wintersteiner and Abramson, 1933**), impairing receptor dissociation and resulting in delayed endosomal maturation with a concomitant entrapment of co-endocytosed glutamate receptors. Endosomal pH is regulated by the vesicular ATPase and the counterregulatory action of the proton leakage channel NHE6. NHE6 is an antiporter that exchanges a $Na^+$ or $K^+$ ion for each proton. As the pH decreases, ligands dissociate from their receptors allowing the EE to mature. If dissociation is delayed, as in case of ApoE4, endosomal trafficking is arrested, leading to progressive acidification as $Na^+$, $K^+$, and $Cl^-$ ions continue to enter the endosome to maintain charge neutrality while also drawing in water molecules due to osmotic pressure. We thus propose a model in which delayed ApoE4-receptor dissociation prevents early endosomal maturation and causes osmotic swelling while the pH continues to decrease until dissociation occurs. (**C**) Accelerated endosomal acidification by inhibition of the proton leak channel NHE6 resolves ApoE4 accumulation, promotes rapid receptor dissociation, and promotes the vesicle entry into the lysosomal delivery or recycling pathways.

(β-CTF) and a soluble extracellular APP domain (sAPPβ). β-CTF is further cleaved by the γ-secretase complex, which leads to the release of the Aβ peptide. APP and its secretases co-localize in endosomal compartments where cleavage can occur (**Wang et al., 2018**). It has further been reported that BACE1 activity is preferentially active in acidic environments (**Shimizu et al., 2008**). We therefore investigated whether NHE6 depletion alters BACE1 activity in neurons and whether NHE6 deficiency leads to changes in plaque deposition in vivo. We found that NHE6 inhibition or knockdown did not alter BACE1 activity, as judged by unchanged Aβ generation. By contrast, NHE6 ablation led to glial activation and decreased plaque load in *Apoe*[APOE4] (**Sullivan et al., 1997**) and *App*[NL-F] (**Saito et al., 2014**) double knockin mice.

## Results

### NHE6 is required for postnatal Purkinje cell survival

NHE6 germline KO mice (*Slc9a6*[-]) and tamoxifen-inducible conditional NHE6 KO mice (*Slc9a6*[fl];*CAG-Cre*[ERT2]) were generated as described in Materials and methods and *Figure 2A–C*. To validate early endosomal pH acidification by NHE6 deficiency, we isolated mouse embryonic fibroblasts from the *Slc9a6*[-] line and infected them with a Vamp3-pHluorin2 lentivirus expressing a fusion protein consisting of the endosomal Vamp3 protein and the ratiometric pH indicator pHluorin2 (**Stawicki et al., 2014**). We found a significantly reduced number of vesicles with pH 6.4 and above in *Slc9a6*[-] fibroblasts when compared to control (*Figure 2D–F*).

To induce the conditional KO (cKO) of *Slc9a6*, tamoxifen was administered to *Slc9a6*[fl];*CAG-Cre*[ERT2] mice at 2 months (*Figure 3A*) and experiments were performed at the indicated time points. *Slc9a6* KO efficiency in the brains of tamoxifen-injected *Slc9a6*[fl];*CAG-Cre*[ERT2] mice was 65–82% and varied between brain regions (*Figure 3B*). To further investigate Cre[ERT2] activity upon tamoxifen injection, we bred the *CAG-Cre*[ERT2] line with a stop-tdTomato reporter line in which a floxed stop-codon precedes the tdTomato start-codon. After tamoxifen injection, brains were examined for tdTomato expression. Without tamoxifen injection, tdTomato-expressing cells were almost absent in the hippocampus. Tamoxifen-induced recombination led to a broad expression of tdTomato in the hippocampus (*Figure 2G*).

Individuals with Christianson syndrome and mice lacking NHE6 present with motor deficits due to a dramatic progressive loss of cerebellar Purkinje cells (**Ouyang et al., 2013**). We have reproduced the Purkinje cell loss in our germline *Slc9a6*[-] mice (*Figure 3C*). Next, we investigated whether Purkinje cell loss is the consequence of neurodevelopmental or neurodegenerative effects caused by loss of NHE6. *Slc9a6* deficiency was induced at 2 months, after Purkinje cells had developed and matured. One year after *Slc9a6* ablation, Purkinje cell loss was indistinguishable from that seen in the germline KO (*Figure 3D–F*). Therefore, Purkinje cell degeneration manifests itself postnatally and is not developmentally determined by the absence of NHE6. However, it is possible that loss of scaffolding functions and proper sorting, rather than dysregulation of endosomal pH, could be the main mechanism that causes Christianson syndrome, including Purkinje cell loss (**Ilie et al., 2020**; **Ilie et al., 2019**). If this could be substantiated by the development or discovery of *Slc9a6* mutants that selectively ablate its ion exchange capacity without affecting its subcellular sorting or interaction with cytoplasmic or

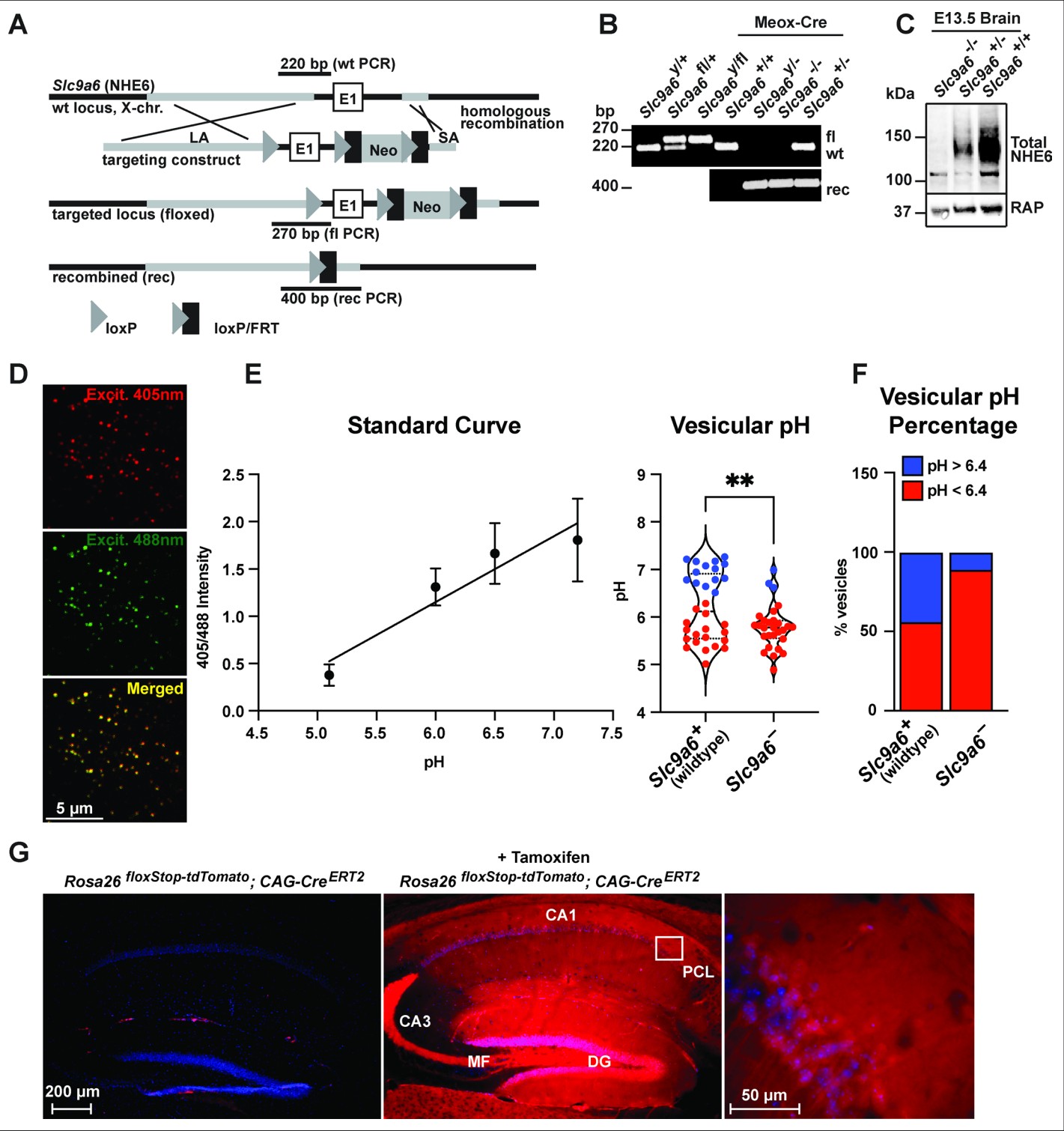

**Figure 2.** Generation of *Slc9a6*<sup>fl</sup> and *Slc9a6*<sup>-</sup> mice. (**A**) Gene targeting strategy. LoxP sites were introduced to flank the first exon (E1) of *Slc9a6* (located on the X-chromosome) by gene targeting in embryonic stem cells. The targeting construct contained a long arm of homology (LA, gray) upstream of the first loxP site and the first exon. An loxP/FRT-flanked neomycin resistance cassette was cloned downstream of the first exon, followed by a short arm of homology (SA, gray). The targeted locus is shown below. Targeted stem cells were used to generate chimeric *Slc9a6*<sup>fl</sup> mice. Germline NHE6 knockout mice (NHE6<sup>-/-</sup> [female], NHE6<sup>y/-</sup> [male]; rec indicates recombined allele) were generated by breeding the *Slc9a6*<sup>fl</sup> line with Meox-Cre mice. (**B**) Genotyping of wildtype (wt, +), floxed (fl), and recombined (rec, -) NHE6 alleles. The PCR amplified regions are indicated in panel A. The wildtype and floxed allele PCR products differ by 50 bp (270 for floxed, 220 for wildtype). (**C**) Western blot showing brain lysates (left) of different NHE6 genotypes after

*Figure 2 continued on next page*

*Figure 2 continued*

Meox-Cre-induced germline recombination. (**D**) Mouse embryonic fibroblasts from *Slc9a6⁻* and control littermate were infected with Vamp3-pHluorin2 and excited at 408 and 488 nm with emission measured at 510 nm. (**E**) Vesicular pH measured using a standard curve was significantly decreased in *Slc9a6⁻* fibroblasts. Data is expressed as mean ± SEM. Statistical analysis was performed using Student's t-test. (**p < 0.01) (**F**) The percent of vesicles with pH >6.4 is significantly decreased in *Slc9a6⁻* fibroblasts. (**G**) *CAG-Cre^ERT2^* activity after tamoxifen application in a reporter mouse line expressing tdTomato. *Cre^ERT2^* recombination activity without (left panel) or with (middle panel) tamoxifen application in the *CAG-Cre^ERT2^* line bred with Rosa26^floxStop-tdTomato^ line. After tamoxifen induction, CreERT2 activity led to a robust tdTomato signal in the hippocampus (middle panel). Pyramidal neurons in the CA1 pyramidal cell layer (PCL) (middle panel) are shown magnified in the right panel.

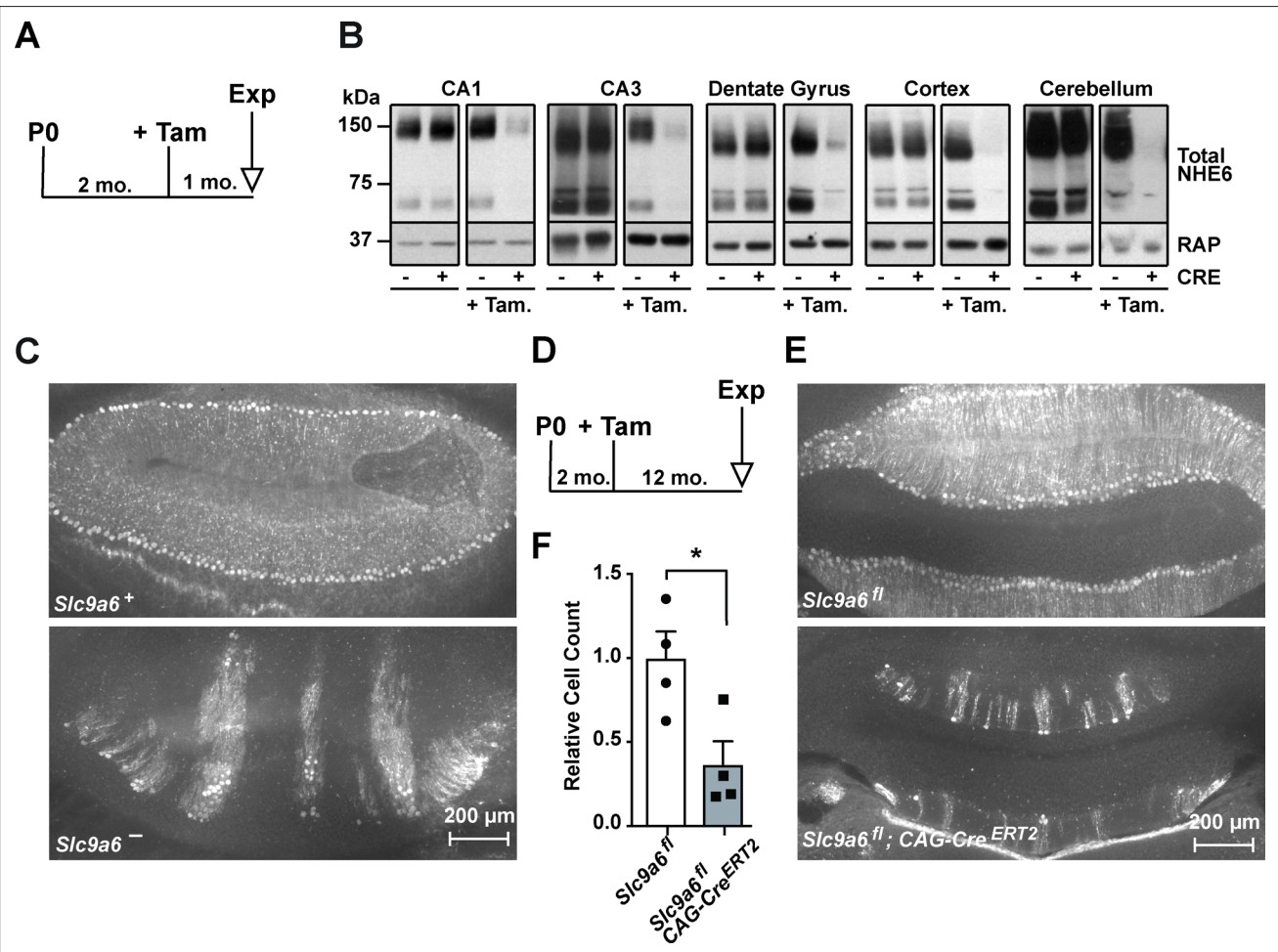

**Figure 3.** Long-term sodium-hydrogen exchanger 6 (NHE6) deficiency induced after Purkinje cell maturation causes Purkinje cell loss. (**A**) Experimental timeline for B, mice were injected with tamoxifen at 2 months; after 1 month the brains were analyzed for NHE6 expression (Tam = tamoxifen, Exp. = experiment, mo. = months). (**B**) Western blot showing the efficiency of tamoxifen-induced NHE6 knockout in different brain regions (CA1, CA3, dentate gyrus, cortex, and cerebellum). The knockout efficiency differed between brain regions, it was 80% ± 2% in CA1, 82 ± 5.7% in the CA3, 67 ± 6.8% in the dentate gyrus, 65% ± 11.2% in the cortex, and 74% ± 4.7% in the cerebellum. A total of three brains in each group were examined. (**C–F**) NHE6 deficiency leads to cerebellar Purkinje cell loss in germline (*Slc9a6⁻*, **C**) and conditional (*Slc9a6^fl^;CAG-Cre^ERT2^*, **D–F**) knockout mice. *Slc9a6⁺* includes both female wildtypes (*Slc9a6^+/+^*) and male wildtypes (*Slc9a6^y/+^*) mice. *Slc9a6⁻* includes both female knockouts (*Slc9a6^-/-^*) and male knockouts (*Slc9a6^y/-^*) mice. In addition, *Slc9a6^fl^* mice includes both female *Slc9a6^fl/fl^* and male *Slc9a6^y/fl^*. The timeline shows that *Slc9a6^fl^;CAG-Cre^ERT2^* and control mice were tamoxifen-injected at 2 months and analyzed 1 year after (**D**). Calbindin was fluorescently labeled to highlight Purkinje cells in the cerebellum. Massive loss of Purkinje cells was found in *Slc9a6⁻* (**C**, lower panel), compared to wildtype *Slc9a6⁺* control (**C**, upper panel). Long-term loss of NHE6, induced after Purkinje cell maturation at 2 months of age, also led to massive Purkinje cells loss when mice were examined 1 year after NHE6 ablation (**E**, lower panel). (**F**) Quantification of Purkinje cell loss in the cerebellum of *Slc9a6^fl^;CAG-Cre^ERT2^* mice. Values are expressed as mean ± SEM from four independent experiments. Statistical analysis was performed using Student's t-test. *p < 0.05.

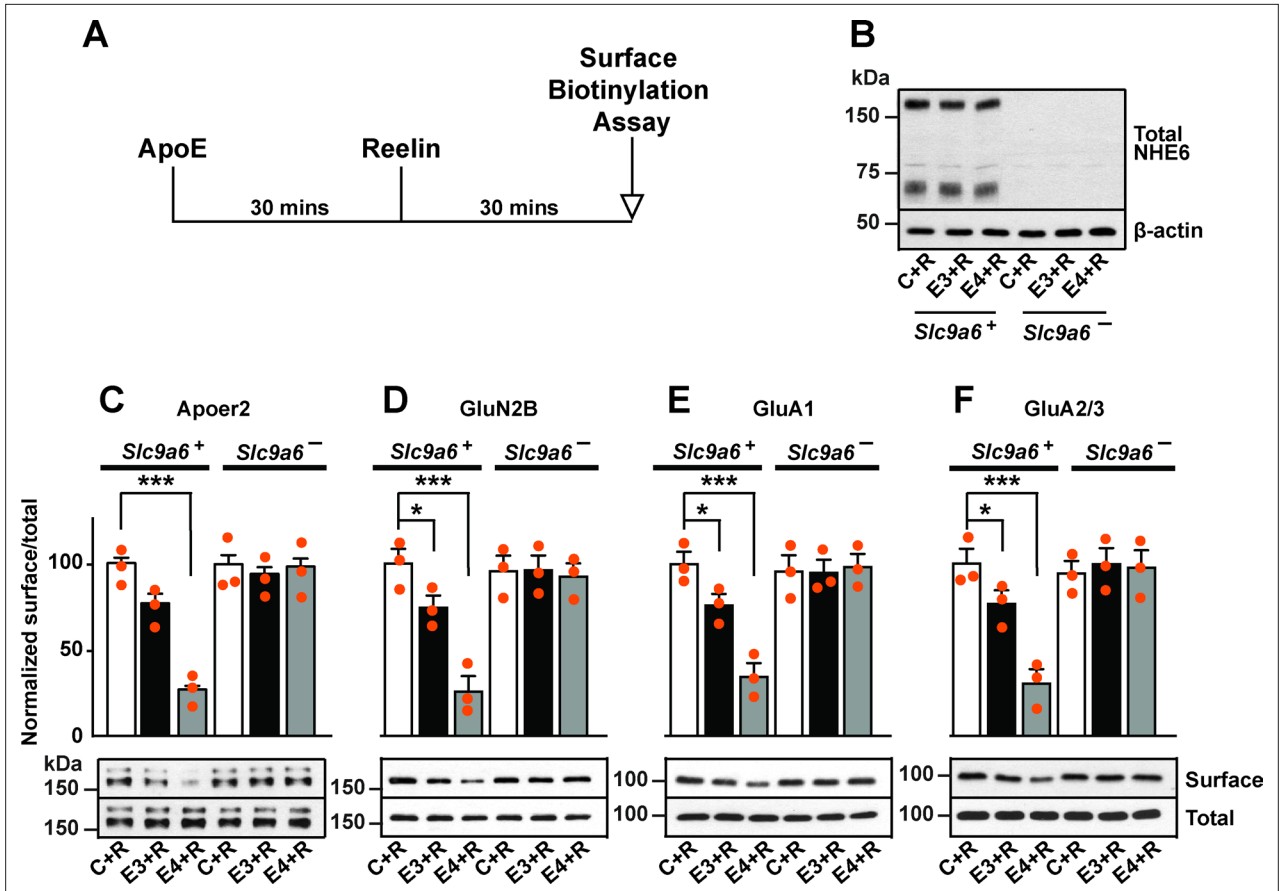

**Figure 4.** Sodium-hydrogen exchanger 6 (NHE6) deficiency alleviates ApoE4-impaired surface trafficking deficits of Apoer2 and glutamate receptors. (**A**) Timeline for the receptor surface expression assay applied for the experiments shown in B–F. Primary neurons were treated with naturally secreted ApoE3 or ApoE4 and/or Reelin before they underwent surface biotinylation. (**B–F**) Wildtype and *Slc9a6*⁻ primary neurons were prepared from littermates and used in the receptor surface expression assay described in A. *Slc9a6*⁺ includes both female wildtypes (*Slc9a6*^{+/+}) and male wildtypes (*Slc9a6*^{y/+}) mice. *Slc9a6*⁻ includes both female knockouts (*Slc9a6*^{-/-}) and male knockouts (*Slc9a6*^{y/-}) mice. (**B**) NHE6 deficiency was confirmed via Western blot, β-actin was used as loading control. (**C–F**) ApoE-conditioned media treatment reduces the surface expression of Apoer2 and glutamate receptors in presence of Reelin in primary neurons. Receptor surface levels show a stronger reduction with ApoE4 than ApoE3. NHE6 depletion counteracts the ApoE4-induced reduction of receptor surface expression. Cell surface biotinylation assay was performed for Apoer2 (**C**), GluN2B (**D**), GluA1, (**E**) and GluA2/3 (**F**). Total levels were analyzed by immunoblotting of whole cell lysates against the same antibodies. β-Actin was used as loading control. Quantitative analysis of immunoblot signals is shown in the lower panels (**C–F**). All data are expressed as mean ± SEM from three independent experiments. *p < 0.05, **p < 0.01, ***p < 0.005. Statistical analysis was performed using one-way analysis of variance (ANOVA) and Dunnett's post hoc test (**C–F**).

luminal binding partners, this would further raise the potential of NHE6 as a novel drug target for neurodegenerative diseases.

## Genetic disruption of NHE6 restores trafficking of Apoer2, AMPA, and NMDA receptors in the presence of ApoE4

As we reported previously, ApoE4 impairs the trafficking of synaptic surface receptors (*Chen et al., 2010*). To monitor receptor recycling in neurons, we made use of an assay where Reelin is used to modulate receptor surface expression. Reelin is applied to primary neurons for 30 min in the presence or absence of naturally secreted, receptor-binding competent ApoE (*Figure 4*). Subsequently, surface biotinylation is performed and cells are harvested for immunoblotting to quantify the amount of Apoer2 and glutamate receptors expressed on the neuronal surface (*Chen et al., 2010*; *Xian et al., 2018*).

We have shown previously that in the presence of ApoE4, Apoer2 and glutamate receptors recycle poorly to the neuronal surface. This recycling block could be resolved by endosomal acidification induced by shRNA knockdown of *Slc9a6* or by applying the NHE inhibitor EMD87580 (*Xian et al.,*

2018). To further exclude a nonspecific effect caused by the inhibition of other NHE family members or by shRNA off-target effects, we applied this assay on neurons isolated from *Slc9a6*⁻ embryos (*Figure 4B*). Apoer2 recycling was completely restored in *Slc9a6*⁻ neurons treated with ApoE4 (*Figure 4C*). We previously reported that the addition of ApoE3 to neurons also affects Apoer2 trafficking to a small, but reproducible extent. This was also abolished in *Slc9a6*⁻ neurons (*Figure 4C*). In addition, genetic loss of NHE6 equally restored the ApoE4-impaired surface expression of AMPA and NMDA receptor subunits (*Figure 4D–F*).

## Conditional disruption of NHE6 relieves synaptic Reelin resistance in *Apoe*^APOE4^ mice

Reelin can enhance long-term potentiation (LTP) in hippocampal field recordings of *Apoe*^APOE3^ but not *Apoe*^APOE4^ acute brain slices (*Chen et al., 2010*). We previously showed that this Reelin resistance in *Apoe*^APOE4^ slices was attenuated by pharmacological NHE inhibition (*Xian et al., 2018*). To investigate if endogenous loss of NHE6 also restores synaptic plasticity in the presence of ApoE4, we performed hippocampal field recordings on *Slc9a6*^fl^;*CAG-Cre*^ERT2^ mice bred to *Apoe*^APOE3^ or *Apoe*^APOE4^ mice. To avoid potentially compounding effects of NHE6 deficiency during embryonic development (*Ouyang et al., 2013*), NHE6/*Slc9a6* gene disruption was induced at 8 weeks by intraperitoneal tamoxifen injection (*Lane-Donovan et al., 2015*). Electrophysiology was performed 3–4 weeks after NHE6 depletion in 3 -month-old mice (*Slc9a6*^fl^;*CAG-Cre*^ERT2^, *Figure 5B, D, F and H*). Tamoxifen-injected *Slc9a6*^fl^;*CAG-Cre*^ERT2^-negative mice expressing human ApoE3 or ApoE4 served as controls (*Apoe*^APOE3^ and *Apoe*^APOE4^, *Figure 5A, C, E and G*). For field recordings, hippocampal slices were perfused with Reelin as described (*Beffert et al., 2005*; *Chen et al., 2010*; *Durakoglugil et al., 2009*; *Weeber et al., 2002*). Conditional genetic loss of NHE6 resulted in a moderate reduction of the ability of Reelin to enhance LTP in *Apoe*^APOE3^ mice (comparing *Figure 5A and E* to B and F). By contrast, as reported previously (*Chen et al., 2010*), hippocampal slices from *Apoe*^APOE4^ mice were completely resistant to LTP enhancement by Reelin (*Figure 5C and G*). This resistance was abolished when NHE6 was genetically disrupted in *Apoe*^APOE4^ mice: Reelin application enhanced LTP (*Figure 5D and H*) in *Apoe*^APOE4^;*Slc9a6*^fl^;*CAG-Cre*^ERT2^ to a comparable extent as in the *Apoe*^APOE3^;*Slc9a6*^fl^;*CAG-Cre*^ERT2^ mice (*Figure 5B*). Synaptic transmission was monitored and input-output curves were generated by plotting the fiber volley amplitude, measured at increasing stimulus intensities, against the fEPSP slope. No significant differences were found (*Figure 5I and J*).

## NHE inhibition or NHE6 knockdown does not alter β-CTF generation in vitro

Cleavage of APP by γ-secretase and the BACE1 generates the short neurotoxic polypeptide Aβ. Cleavage by BACE1 results in a membrane anchored fragment called β-CTF, which is further processed by γ-secretase to yield the soluble Aβ peptide. BACE1 processing of APP occurs in the Golgi complex, on the cell membrane, and after endocytosis in endosomes (*Caporaso et al., 1994*; *Vassar et al., 1999*). It has been reported that BACE1 activity increases with lower pH (*Hook et al., 2002*). In a recent in vitro study in an HEK293 cell line overexpressing APP and BACE1, NHE6 overexpression reportedly led to a reduction of Aβ production and conversely shRNA knockdown of NHE6/*Slc9a6* resulted in an increase in Aβ production (*Prasad and Rao, 2015*). To investigate if NHE6/*Slc9a6* deficiency contributes to APP processing by BACE1 in neurons, we used primary neurons derived from *AppSwe* (*Tg2576*) mice, an Alzheimer's disease (AD) mouse model that overexpresses human APP with the 'Swedish' mutation (*Hsiao et al., 1996*). Neurons were treated with the NHE inhibitor EMD87580 or transduced with lentiviral shRNA directed against NHE6/*Slc9a6* in the presence or absence of the γ-secretase inhibitor L-685458. β-CTF was detected using the monoclonal antibody 6E10 directed against Aβ residues 1–16 (*Figure 6*). Inhibition of γ-secretase in ApoE4-treated *AppSwe* neurons strongly enhanced β-CTF accumulation, however, additional treatment with EMD87580 did not alter the amount of β-CTF in the cell lysates (*Figure 6A*). NHE6 knockdown using lentiviral shRNA also had no effect on the amount of β-CTF (*Figure 6B*). We conclude that NHE6 inhibition is unlikely to increase Aβ production under near-physiological conditions.

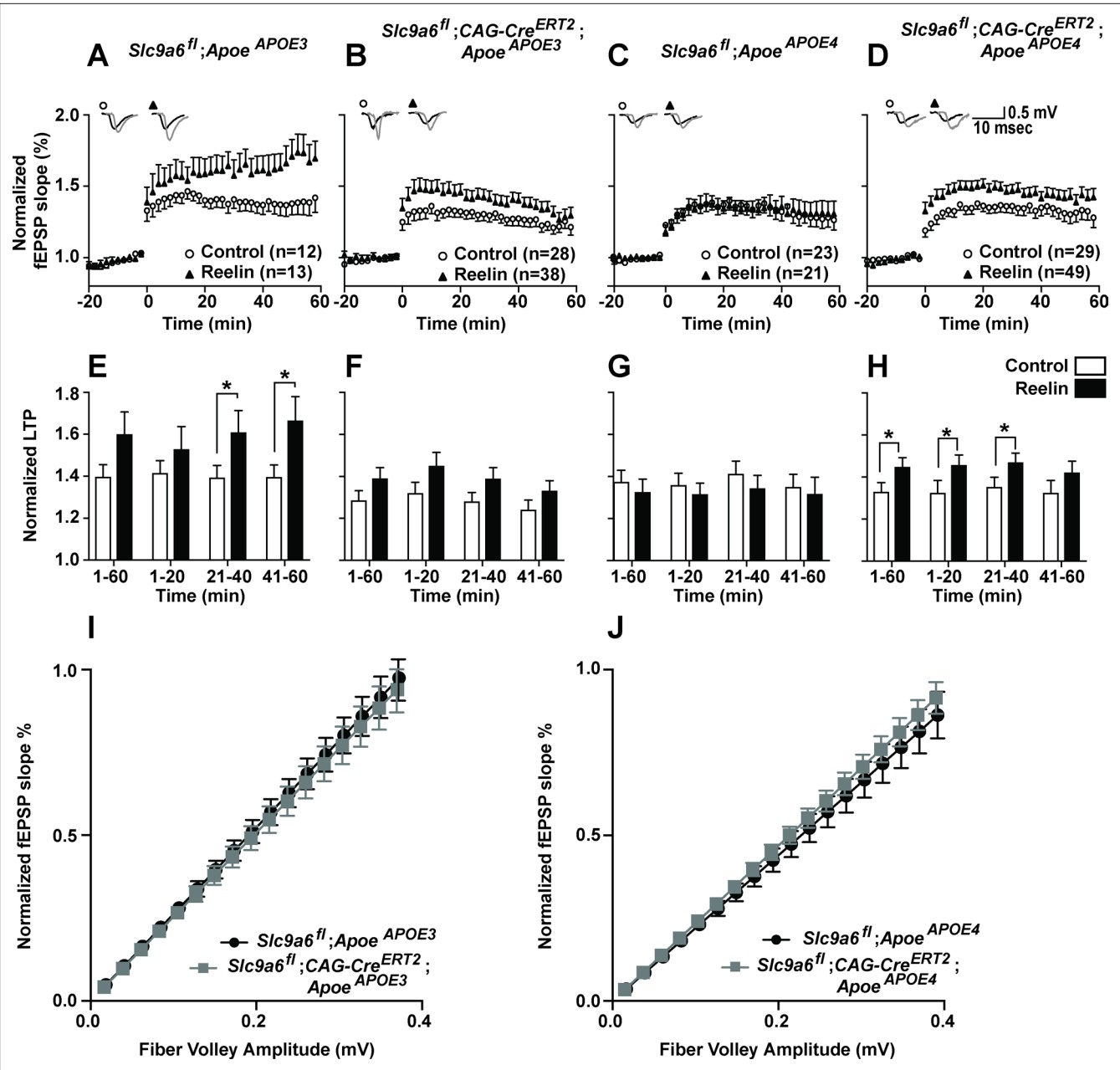

**Figure 5.** Effect of conditional sodium-hydrogen exchanger 6 (NHE6) knockout on Reelin-potentiated synaptic plasticity. (**A–H**) Conditional knockout of NHE6 restores Reelin-enhanced long-term potentiation (LTP) in $Apoe^{APOE4}$ mice. Reelin facilitated induction of LTP in $Apoe^{APOE3}$ (**A, E**), but not $Apoe^{APOE4}$ (**C, G**) control ($Slc9a6^{fl}$) mice. $Slc9a6$ deficiency in $Apoe^{APOE3}$ mice caused a reduction in Reelin-enhanced LTP, such that it is not significantly different from the control LTP (**B, F**). Importantly, in $Apoe^{APOE4}$;$Slc9a6^{fl}$;$CAG-Cre^{ERT2}$ mice Reelin was able to enhance theta burst-induced potentiation (**D, H**). Hippocampal slices were prepared from 3 -month-old double mutant mice with either human $Apoe^{APOE3}$ or $Apoe^{APOE4}$ crossed with $Slc9a6$ conditional knockout mice ($Slc9a6^{fl}$;$CAG-Cre^{ERT2}$, tamoxifen injections at 6–8 weeks). Extracellular field recordings were performed in slices treated with or without Reelin. Theta burst stimulation (TBS) was performed after 20 min of stable baseline. Representative traces are shown in each panel, before TBS induction (black) and 40 min after TBS (gray). (**E–H**) Quantitative analysis of normalized fEPSP slopes at time intervals as indicated. (**I, J**) Input output curves of $Apoe^{APOE3}$ (**I**) and $Apoe^{APOE4}$ (**J**) mice with or without $Slc9a6^{fl}$;$CAG-Cre^{ERT2}$. $Slc9a6^{fl}$ mice includes both female $Slc9a6^{fl/fl}$ and male $Slc9a6^{y/fl}$ mice. $Apoe$ mice are homozygous for APOE3 or APOE4. All data are expressed as mean ± SEM. N-numbers for each genotype group and treatment are indicated in panels A–D. *p < 0.05. Statistical analysis was performed using Student's t-test.

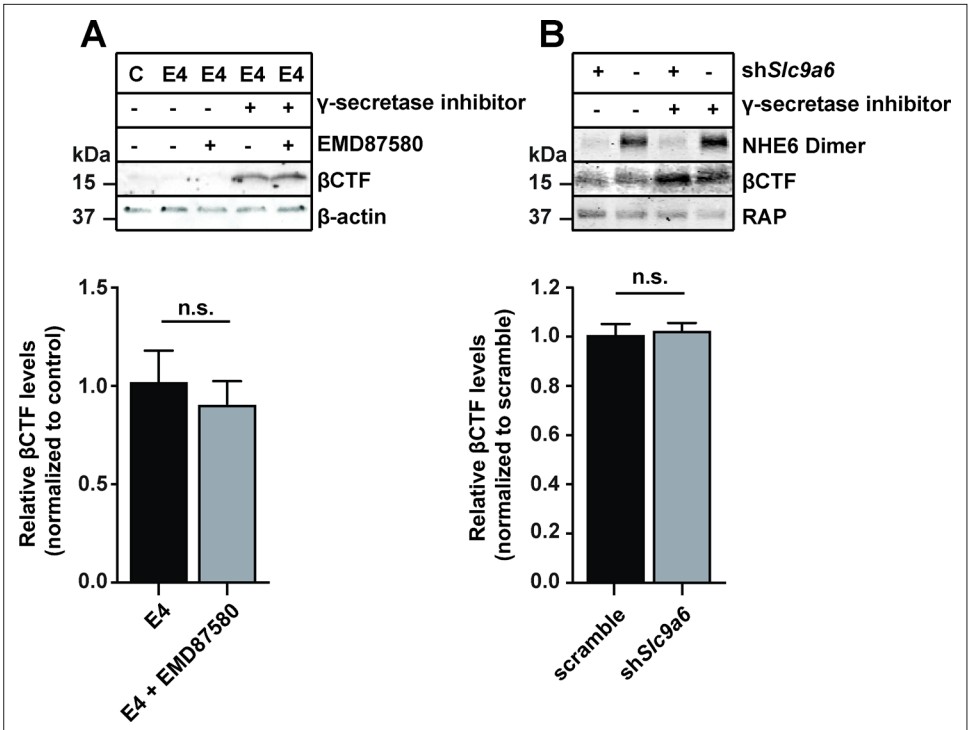

**Figure 6.** Na⁺/H⁺ exchanger (NHE) inhibition or sodium-hydrogen exchanger 6 (NHE6) knockdown does not alter β-site amyloid precursor protein cleaving enzyme 1 (BACE1) activity in primary neurons. (**A, B**) Pan-NHE inhibition by EMD87580 or lentiviral knockdown of *Slc9a6* (NHE6) did not alter BACE1 activity in primary neurons of *AppSwe* mice (Tg2576). (**A**) DIV10 primary neurons were treated with γ-secretase inhibitor L-685458, EMD87580, and/or ApoE4 (as indicated) and harvested for immunoblotting against Aβ-containing C-terminal fragment of APP (β-CTF). β-Actin was blotted as loading control. Bar graph shows the statistics of n = 3 experiments. (**B**) Primary neurons of *AppSwe* mice were infected with lentivirus for shRNA expression directed against *Slc9a6*(NHE6) (sh*Slc9a6*) or a scramble control sequence (-) at DIV7. At DIV13 neurons were treated with L-685458 overnight and harvested for immunoblotting against NHE6 and β-CTF on DIV14. RAP was blotted as loading control. Bar graph shows the statistics of n = 6 experiments. All data are expressed as mean ± SEM. Statistical analysis was performed using Student's t-test. n.s. = not significant.

## NHE6 deficiency reduces Aβ plaque load in human *App* knockin mice

To further study the effect of NHE6 deficiency on Aβ pathology in vivo, we bred humanized *App^{NL-F}* mice (*Saito et al., 2014*) to our germline NHE6 KO line (*Slc9a6^-*). In these *App^{NL-F}* mice, the Aβ sequence has been completely humanized and the early onset AD Swedish mutation (5' located mutations encoding K670N and M671L = NL) and the Beyreuther/Iberian mutation (3' flanking mutation encoding I716*F* = F) were also introduced, resulting in increased Aβ production, but physiological regulation of APP expression. This allowed us to determine the effect of Aβ overproduction while keeping APP expression under the control of the endogenous promoter. *App^{NL-F};Slc9a6^-* and control *App^{NL-F}* littermates were aged to 1 year. Perfusion-fixed brains were harvested and analyzed by hematoxylin and eosin (H&E) staining, Thioflavin S staining to visualize plaque load, and Aβ immunohistochemistry. H&E staining did not reveal any obvious anatomical structural differences between genotypes, but brain size, cortical thickness, hippocampal area, and CA1 thickness were reduced, as described previously (*Xu et al., 2017*; *Figure 7—figure supplement 1*). Plaques were more frequent in *Slc9a6 wildtype* than *Slc9a6^-* mice. To further investigate and quantify plaque load, we analyzed the same brains after Thioflavin S staining (*Figure 7A*). We found an approximate 80 % reduction of plaques in *Slc9a6^-* mice when compared to littermate controls (*Figure 7B*). Immunohistochemistry against Aβ showed the same reduction (*Figure 7—figure supplement 3A,B*). In addition, we analyzed soluble (TBS fraction) and insoluble (GuHCl and 70 % FA fractions) Aβ in cortical brain lysates of 1.5 -year-old *App^{NL-F};Slc9a6^-* mice and their control littermates. The amount of insoluble Aβ (GuHCl and 70 % FA fractions) was reduced in NHE6-depleted mice by approximately 71%, when compared

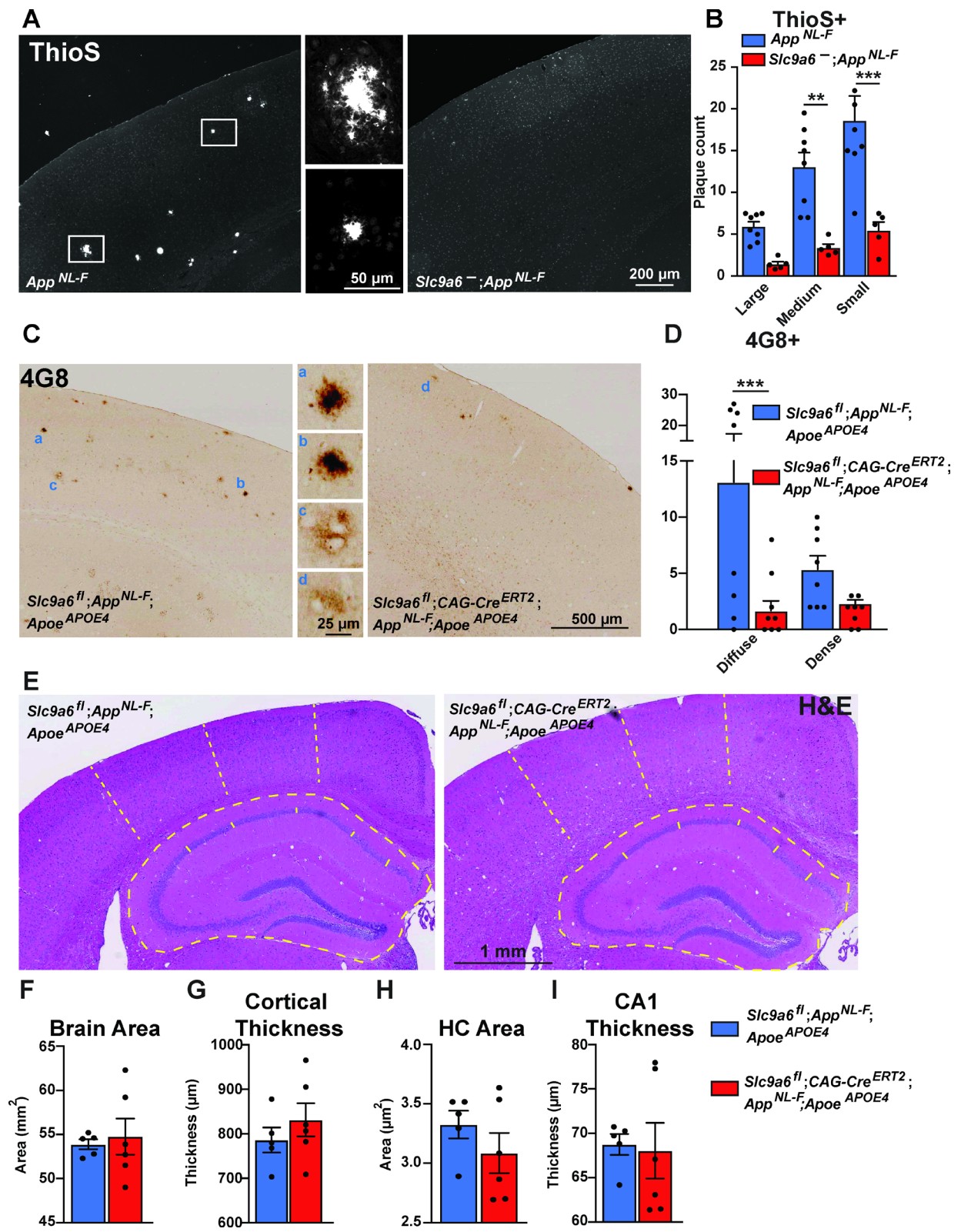

**Figure 7.** Sodium-hydrogen exchanger 6 (NHE6) deficiency decreases plaque formation in both *App*[NL-F] and *App*[NL-F];*Apoe*[APOE4] mice. (**A, B**) NHE6-deficient *App*[NL-F] and control *App*[NL-F] mice were analyzed for plaque deposition at an age of 12 months. Thioflavin S staining was performed to visualize plaques. Plaques were found more frequently in the control *App*[NL-F] mice (left panel in **A**), magnifications of the boxed areas are shown in the two middle panels. The plaque load between *Slc9a6*[-] mice and control littermates (all *App*[NL-F]) was compared and analyzed. (**B**) In the *Slc9a6*[-] littermates, the plaque

*Figure 7 continued on next page*

*Figure 7 continued*

number was reduced, when compared to controls. (**C–D**) *Slc9a6*$^{fl}$*;CAG-Cre*$^{ERT2}$*;App*$^{NL-F}$*;Apoe*$^{APOE4}$ and *Slc9a6*$^{fl}$*;App*$^{NL-F}$*;Apoe*$^{APOE4}$ mice were analyzed for plaque deposition. NHE6 was ablated at 2 months and brains were analyzed at 13.5–16 months. 4G8-immunolabeling against Aβ was performed to visualize plaques. In *App*$^{NL-F}$*;Apoe*$^{APOE4}$ mice conditional *Slc9a6* knockout caused a reduction in plaque load compared to the *Slc9a6*$^{fl}$ control littermates. (**C**) Magnifications of the boxed areas in C are shown in the middle. (**D**) Plaque load was analyzed and compared between *Slc9a6*$^{fl}$*;CAG-Cre*$^{ERT2}$ mice and floxed control littermates. (**I**) Hematoxylin and eosin (H&E) staining was performed to investigate for gross anatomic abnormalities in the *Slc9a6*$^{fl}$*;CAG-Cre*$^{ERT}$*;App*$^{NL-F}$*;Apoe*$^{APOE4}$ and *Slc9a6*$^{fl}$*;App*$^{NL-F}$*;Apoe*$^{APOE4}$ mice. (**F–I**) Brain area (**F**), cortical thickness (**G**), hippocampal (HC) area (**H**), and CA1 thickness (**I**) were analyzed. Student's t-test did not reveal a significant difference. Plaques were differentiated by size or staining density as described in detail in the supplements (*Figure 7—figure supplement 2*). Labeled plaques were analyzed by a blinded observer. All data are expressed as mean ± SEM. (**B**) *Slc9a6*$^-$ n = 5, control n = 8, (**C**) *Slc9a6*$^{fl}$ n = 8 (*Slc9a6*$^{fl}$*;CAG-Cre*$^{ERT2}$ n = 8), in (**F–I**) derived from n = 5 (*Slc9a6*$^{fl}$) and n = 6 (*Slc9a6*$^{fl}$*;CAG-Cre*$^{ERT2}$) animals. *p < 0.05. **p < 0.01, ***p < 0.005. *Slc9a6*$^+$ represents both female wildtypes (*Slc9a6*$^{+/+}$) and male wildtypes (*Slc9a6*$^{y/+}$). *Slc9a6*$^-$ represents both female knockouts (*Slc9a6*$^{-/-}$) and male knockouts (*Slc9a6*$^{y/-}$). In addition, *Slc9a6*$^{fl}$ mice includes both female *Slc9a6*$^{fl/fl}$ and male *Slc9a6*$^{y/fl}$ mice. *Apoe* mice are homozygous for APOE4 (*Apoe*$^{APOE4}$). *App*$^{NL-F}$ mice are homozygous for human NL-F knockin mutation (*App*$^{NL-F/NL-F}$). Statistical analysis was performed using two-way analysis of variance (ANOVA) with Sidak's post hoc test (**B and D**) and Student's t-test (**F–I**).

The online version of this article includes the following figure supplement(s) for figure 7:

**Figure supplement 1.** Gross anatomical brain structure in *Slc9a6*$^-$ mice.

**Figure supplement 2.** Example of Thioflavin S stained plaques for quantification.

**Figure supplement 3.** Sodium-hydrogen exchanger 6 (NHE6) deficiency decreases plaque formation in both *App*$^{NL-F}$ and *App*$^{NL-F}$*;Apoe*$^{APOE4}$ mice.

---

to their control littermates (*Figure 7—figure supplement 3C-E*). The ~50 % reduction in soluble Aβ was statistically not significant in *Slc9a6*$^-$ lysates (TBS fraction).

## NHE6/*Slc9a6* deficiency reduces plaque load in *App*$^{NL-F}$;*Apoe*$^{APOE4}$

To further investigate whether *Slc9a6* deficiency also protects the brain from plaques in the presence of human ApoE4 instead of murine ApoE, we bred *Slc9a6*$^{fl}$*;CAG-Cre*$^{ERT2}$*; Apoe*$^{APOE4}$ with *App*$^{NL-F}$ mice. At 2 months of age, we induced *Slc9a6* ablation with tamoxifen and aged the mice to 14–16 months. *Slc9a6* deficiency on the background of human ApoE4 reduced plaque deposition, as shown by Thioflavin S staining (*Figure 7—figure supplement 3F, G*) and 4G8 immunoreactivity (*Figure 7C and D*). The age-dependent increase in plaque load in *Apoe*$^{APOE4}$*;App*$^{NL-F}$ mice was abolished or delayed when *Slc9a6* was depleted after 2 months of age (*Figure 8*). *App*$^{NL-F}$ mice expressing murine ApoE developed plaques at 12 months (*Figure 7A and B*), compared to *Apoe*$^{APOE4}$*;App*$^{NL-F}$ which showed a similar number of plaques at 15–16 months (*Figures 7C–D and 8*, and *Figure 7—figure supplement 3F-G*). This delay of plaque deposition caused by the presence of human ApoE4 as opposed to murine ApoE is consistent with earlier findings by the Holtzman group (*Liao et al., 2015*). Importantly, NHE6/*Slc9a6* ablation induced at 2 months showed a comparable reduction of plaque load as germline NHE6/*Slc9a6* depletion. We conclude that plaque deposition is modulated by the presence of NHE6 postnatally and is not affected by NHE6 activity during development.

## NHE6/*Slc9a6* deficiency does not affect cortical thickness and hippocampus size in *App*$^{NL-F}$;*Apoe*$^{APOE4}$ cortices

*Xu et al., 2017*, reported reduced cortex thickness and hippocampus volume in *Slc9a6*$^-$ mice, which we were able to reproduce in our germline *Slc9a6*$^-$ model (*Figure 7—figure supplement 1*). In addition, based on their findings *Xu et al., 2017*, concluded that the difference in brain size was a combined result of both neurodevelopmental and neurodegenerative effects caused by NHE6/*Slc9a6* deficiency. Since both germline deficiency and adult-onset deficiency of NHE6 causes massive Purkinje cell loss in the cerebellum (*Figure 3C–E*), we next investigated the effect of conditional NHE6/*Slc9a6* loss on hippocampal and cortical neuronal loss in our *App*$^{NL-F}$*;Apoe*$^{APOE4}$ model (*Figure 7E–I*). We measured brain size, cortical thickness, hippocampal area, and CA1 thickness. In contrast to germline *Slc9a6*$^-$ mice (*Figure 7—figure supplement 1*), none of the analyzed parameters differed significantly between *Slc9a6*$^{fl}$*;CAG-Cre*$^{ERT2}$ and controls (*Figure 7E–I*), and we specifically did not detect any reduction in brain size compared to *Slc9a6*$^{fl}$*;App*$^{NL-F}$*;Apoe*$^{APOE4}$ littermate controls. This is consistent with the undergrowth model proposed by *Xu et al., 2017*, as we induced conditional disruption of NHE6/*Slc9a6* in the adult after postnatal brain growth was completed.

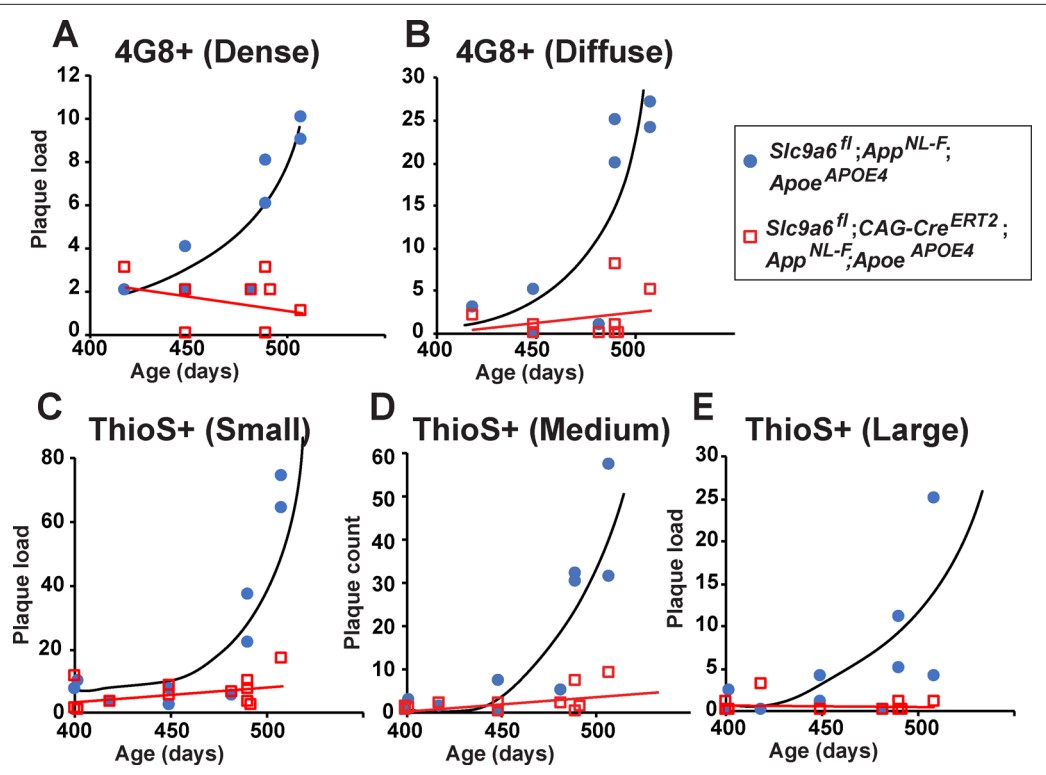

**Figure 8.** Age-dependent increase in plaque load is abolished in *Slc9a6*<sup>fl</sup>;*CAG-Cre*<sup>ERT2</sup>;*App*<sup>NL-F</sup>;*Apoe*<sup>APOE4</sup> mice. (**A–E**) *Slc9a6*<sup>fl</sup>;*CAG-Cre*<sup>ERT2</sup>;*App*<sup>NL-F</sup>;*Apoe*<sup>APOE4</sup> and *Slc9a6*<sup>fl</sup>;*App*<sup>NL-F</sup>;*Apoe*<sup>APOE4</sup> mice were analyzed for plaque deposition. Sodium-hydrogen exchanger 6 (NHE6) was ablated at 2 months and brains were analyzed at 13.5–16 months. 4G8-immunolabeling against Aβ (**A,B**) and Thioflavin S staining (**C–E**) were performed to visualize plaques (*Figure 7C* and *Figure 7—figure supplement 3F*). Plaque load was analyzed and compared between *Slc9a6*<sup>fl</sup>;*CAG-Cre*<sup>ERT2</sup> mice and floxed control littermates. Plaques were differentiated by staining intensity (**A, B**) or size (**C–E**) as described in the supplements (*Figure 7—figure supplement 2*). In the time range analyzed, plaque load increased by age in control, but not in *Slc9a6*<sup>fl</sup>;*CAG-Cre*<sup>ERT2</sup> mice. Plaques were analyzed by a blinded observer. Plaque count (*Slc9a6*<sup>fl</sup>;*CAG-Cre*<sup>ERT2</sup> n = 8, *Slc9a6*<sup>fl</sup> n = 8 for **A**) (**B**); *Slc9a6*<sup>fl</sup>;*CAG-Cre*<sup>ERT2</sup> n = 12; *Slc9a6*<sup>fl</sup> n = 10 in (**C–E**) is pis plotted against age of mice. *Slc9a6*<sup>fl</sup> mice includes both female *Slc9a6*<sup>fl/fl</sup> and male *Slc9a6*<sup>y/fl</sup> mice. *Apoe* mice are homozygous for APOE4. *App*<sup>NL-F</sup> mice are homozygous for human NL-F knockin mutation (*App*<sup>NL-F/NL-F</sup>).

## NHE6 deficiency increases Iba1 and GFAP expression in the brain

Neuroprotective astrocytes and microglia have been described to reduce Aβ deposition in early stages of AD (*Sarlus and Heneka, 2017*). It has been reported that NHE6 deficiency leads to increased glial fibrillary acidic protein (GFAP) and ionized calcium-binding adapter molecule (Iba1) immunoreactivity in different brain regions (*Xu et al., 2017*). To validate if plaque load correlated with Iba1 and/or GFAP immunoreactivity, we performed DAB immunostaining for both marker proteins. We found that Iba1 and GFAP immunoreactivity is increased in the white matter and to a lesser extent in the cortex of *App*<sup>NL-F</sup>;*Slc9a6*<sup>-</sup> (*Figure 9—figure supplement 1A, C, D*, C, and D) and *App*<sup>NL-F</sup>;*Apoe*<sup>APOE4</sup>;*Slc9a6*<sup>fl</sup>;*CAG-Cre*<sup>ERT2</sup> (*Figure 9—figure supplement 1B, E, F*) mice compared to their littermate controls. There was a non-significant trend toward increased immunoreactivity for both markers in the hippocampus of the *Slc9a6*<sup>-</sup> group. Taken together, these data are consistent with the findings of the Morrow group in germline *Slc9a6*<sup>-</sup> mice (*Xu et al., 2017*). Neurodegeneration can be a trigger for glial activation (*Yanuck, 2019*). However, our data on glial activation and cerebral volume in cKO mice, where NHE6/*Slc9a6* was disrupted postnatally, suggest that neurodegeneration induced by NHE6/*Slc9a6* deficiency is unlikely to be the trigger for the glial activation we observe.

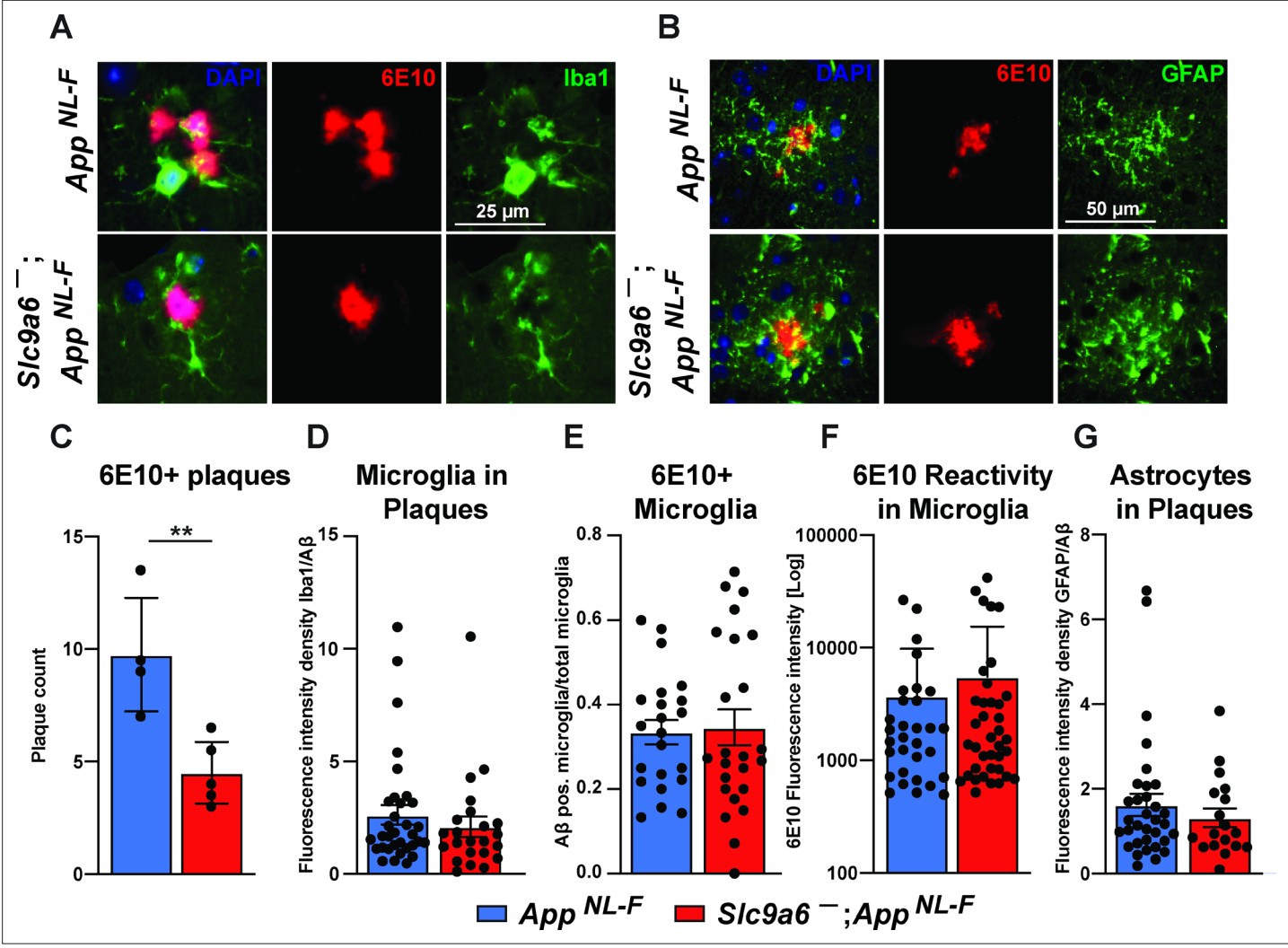

**Figure 9.** Microglia and astrocytes surround plaques in both *App^NL-F* control and *App^NL-F;Slc9a6^-* brains. (**A–B**) Co-labeling of microglia (Iba1, green, **A**) or astrocytes (GFAP, green, **B**) with Aβ (6E10, red) in brain slices of *App^NL-F* and *App^NL-F;Slc9a6^-* mice. (**C**) Quantification of plaques in control and *Slc9a6^-* brain slices. (**D**) Bar graph showing the intensity density of Iba1/6E10 as quantitative measure of microglia surrounding plaques. (**E**) Statistical analysis of 6E10 positive microglia and (**F**) the intensity of 6E10 signal within microglia. (**G**) Bar graph showing the intensity density of GFAP/Aβ as quantitative measure of astrocytes surrounding plaques. Data were analyzed by a blinded observer. All data are expressed as mean ± SEM. Data were obtained from n = 4 (control) and n = 5 (*Slc9a6^-*) mice (**A–G**). (**D**) n = 33 (control) and n = 23 (*Slc9a6^-*) plaques were analyzed, in (**E**) n = 22 (control) and n = 24 (*Slc9a6^-*) microscopical pictures were analyzed, in (**F**) n = 31 (control) and n = 38 (*Slc9a6^-*) 6E10 positive (defined as signal intensity above 500) microglia were analyzed, in (**G**) n = 33 (control) and n = 18 (*Slc9a6^-*) plaques were analyzed. Student's t-test revealed a difference in **C** (**p < 0.01) and did not reveal significant differences in (**D–G**). *Slc9a6^+* represents both female wildtypes (*Slc9a6^+/+*) and male wildtypes (*Slc9a6^y/+*). *Slc9a6^-* represents both female knockouts (*Slc9a6^-/-*) and male knockouts (*Slc9a6^y/-*). *App^NL-F* mice are homozygous for human NL-F knockin mutation (*App^NL-F/NL-F*).

The online version of this article includes the following figure supplement(s) for figure 9:

**Figure supplement 1.** Sodium-hydrogen exchanger 6 (NHE6) deficiency causes an increase in Iba1 and glial fibrillary acidic protein (GFAP) immunoreactivity in both *App^NL-F* and *App^NL-F;Apoe^APOE4* mice.

**Figure supplement 2.** Examples of plaques surrounded by microglia and astrocytes.

## Microglia and astrocytes surround plaques in both *slc9a6^-* and control *App^NL-F* mice

Conditional and germline *Slc9a6* deficient *App^NL-F* mice have a reduced number of plaques in the brain and an increase of Iba1 and GFAP labeled glia. Others have shown that Aβ plaques levels are reduced with increased plaque-associated microglia detected with a co-stain for Iba1 and Aβ (*Parhizkar et al., 2019*; *Zhong et al., 2019*). In order to investigate the contribution of microglia and astrocytes to the

observed plaque reduction, we analyzed brain slices by co-staining for Aβ using the 6E10 antibody (*Figure 9* and *Figure 9—figure supplement 2*). Whereas the total amount of plaques labeled by 6E10 was reduced in *Slc9a6⁻* as compared to control mice (*Figure 9C*), the amount of microglia and astrocytes surrounding plaques did not differ between genotypes (*Figure 9D and G*). In addition, there was no difference between genotypes in the amount of microglia co-labeled with 6E10 (*Figure 9E*) or the intensity of immunoreactivity for 6E10 within microglia (*Figure 9F*).

## Discussion

The prevalence of AD is increasing with life expectancy in all human populations. ApoE4 is the most important genetic risk factor. This makes it of paramount importance to understand the underlying mechanisms by which ApoE4 contributes to the pathology of the disease in order to devise effective targeted therapies that can be deployed on a global scale. Only small molecule drug therapies or, alternatively, immunization approaches can satisfy this requirement. Biologics, including monoclonal antibodies and potential viral gene therapy approaches, are unlikely to be sufficiently scalable. We have previously reported a novel small molecule intervention that has the potential to neutralize ApoE4 risk (*Xian et al., 2018*) through prevention of the ApoE4-induced endosomal trafficking delay of synaptic receptors by the early endosomal sorting machinery. The mechanistic basis of this conceptually novel intervention is the acidification of the early endosomal compartment through inhibition of NHE6. Remarkably and unexpectedly, loss of NHE6/*Slc9a6* effectively reduced Aβ accumulation even in the absence of ApoE4, suggesting that hyperacidification of EEs prevents amyloid plaque formation independently of ApoE4. NHE6/*Slc9a6* suppression or inhibition may thus be a universal approach to prevent amyloid buildup in the brain, irrespective of ApoE genotype. Our previous (*Xian et al., 2018*) and current studies thus suggest a novel mechanism to prevent ApoE4-risk for AD and delay plaque formation. In addition, genome-wide association studies (GWAS) in conjunction with studies on cell culture and mouse models of AD show that various AD risk factors enhance endolysosomal dysfunction (*Knopman et al., 2021*; *Small et al., 2017*; *Verheijen and Sleegers, 2018*), which potentially could be corrected by NHE6 inhibition.

Upon endocytosis, endosomes undergo gradual acidification controlled by vATPases which actively pump protons into the vesicular lumen, and by the Na⁺/H⁺ exchanger NHE6 which functions as a regulatable proton leak channel. NHE6 depletion acidifies early and recycling endosomes (*Lucien et al., 2017*; *Ohgaki et al., 2010*; *Ouyang et al., 2013*; *Xinhan et al., 2011*). pH is an important regulator of the endolysosomal sorting machinery in which vesicles undergo multiple rounds of fusion. EEs undergo fusion and fission events in close proximity to the cell membrane. Recycling endosomes originate from EEs while they undergo early-to-late endosomal maturation. In contrast to late endosomes, recycling endosomes do not undergo further acidification (*Jovic et al., 2010*; *Schmid, 2017*). The pH of EEs and recycling endosomes is ~6.4–6.5 (*Casey et al., 2010*). This is normally sufficient to induce ligand receptor dissociation and enable cargo sorting. Our data, however, suggest that ApoE4 dramatically delays this fast recycling step in neurons, where ApoE, Apoer2, and glutamate receptors co-traffic through fast recycling compartments upon Reelin stimulation (*Xian et al., 2018*). We have proposed that ApoE4 impairs vesicle recycling due to isoelectric precipitation and structural unfolding at the physiological pH of the EE environment. This delays the dissociation of ApoE4 from its receptors, which in turn prolongs the entry of ApoE4 – along with Apoer2 and glutamate receptors in the same vesicle – into the recycling pathway (illustrated in *Figure 1*). We conclude that ApoE4 net charge affects its endosomal trafficking. This is further supported by recent findings by Arboleda-Velasquez and colleagues (*Arboleda-Velasquez et al., 2019*), who reported the presence of the 'Christchurch' R136S mutation in an E3/E3 PS1 mutation carrier without dementia. By neutralizing the positive charge of Arg136 in ApoE3, the IEP of this ApoE3 isoform is predicted to match that of ApoE2, which is protective against AD (*Corder et al., 1994*). ApoE2 homozygous carriers have an exceptionally low likelihood of developing AD (*Reiman et al., 2020*). Moreover, the Christchurch mutation is located within the heparin-binding domain of ApoE, which reduces its affinity for cell surface heparan sulfate proteoglycans. That in turn would result in decreased uptake and thus depletion of ApoE in EEs. The net effect would be unimpeded trafficking of EE vesicles through the fast recycling compartment.

In dendritic spines, NHE6 co-localizes with markers of early and recycling endosomes and with the glutamate receptor subunit GluA1. In the hippocampus, NHE6 is highly expressed in the pyramidal cells of the CA and the granule cells of the dentate gyrus (*Strømme et al., 2011*). Apoer2 is present

at the postsynaptic density of CA1 neurons (*Beffert et al., 2005*). During LTP induction, translocation of NHE6-containing vesicles to dendritic spine heads is enhanced (*Deane et al., 2013*) and glutamate receptors are recruited to the synaptic surface through fast recycling (*Fernández-Monreal et al., 2016*). Our findings are consistent with a model where NHE6 serves as a pH regulator of Reelin-controlled fast recycling endosomes containing Apoer2 and glutamate receptors. This mechanism possibly translates to other cell types and other ApoE receptors.

We previously showed that ApoE4 impairs Reelin-mediated receptor recruitment to the neuronal surface and this can be reversed by functionally disabling NHE6 in primary neurons, which results in the increased acidification of EEs to a level sufficiently different from the IEP of ApoE4, which then allows its efficient dissociation from its receptors. Conditional *Slc9a6* deletion accordingly alleviates the ApoE4-mediated resistance to Reelin-enhanced synaptic plasticity in hippocampal field recordings (*Gao et al., 2019*; *Xian et al., 2018*).

Aβ and tau, forming amyloid plaques and neurofibrillary tangles, are the defining features of AD pathology. As of today, it remains controversial how ApoE isoforms interfere with Aβ and tau pathology. ApoE, which is primarily expressed by astrocytes, is the major lipid transporter in the brain and in an isoform-dependent manner affects inflammatory, endolysosomal, and lipid-metabolic pathways (*Gao et al., 2018*; *Minett et al., 2016*; *Van Acker et al., 2019*; *Xian et al., 2018*). Most risk factors identified by GWAS, including but not limited to APOE, ABCA7, CLU, BIN1, TREM2, SORL1, PICALM, CR1, are members of one or more of these pathways (*Kunkle et al., 2019*). In recent years, endosomal dysfunction has increasingly gained acceptance as a causal mechanism for late-onset AD. Our findings now provide a mechanistic explanation how ApoE4 impairs endolysosomal trafficking and recycling, by interfering with vesicular sorting and maturation at a crucial bottleneck juncture of the endosomal trafficking machinery. This has far-reaching consequences for neuronal function, synaptic plasticity, and tau phosphorylation (*Brich et al., 2003*; *Cataldo et al., 2000*; *Chen et al., 2005*; *Chen et al., 2010*; *Nuriel et al., 2017*; *Pensalfini et al., 2020*). More specifically, ApoE4 causes abnormalities of Rab5-positive endosomes (*Nuriel et al., 2017*). Intriguingly, over-activation of the small guanosine triphosphatase (GTPase) Rab5, recapitulates neurodegenerative features of AD (*Pensalfini et al., 2020*).

ApoE4 alters APP processing and Aβ degradation (reviewed in *Benilova et al., 2012*; *Haass et al., 2012*; *Huynh et al., 2017*; *Lane-Donovan and Herz, 2017*; *Pohlkamp et al., 2017*; *Yamazaki et al., 2019*), and the ability of Reelin to protect the synapse from Aβ toxicity is impaired by ApoE4 (*Durakoglugil et al., 2009*). Aβ oligomerization followed by plaque formation is one hallmark of AD. NHE6 controls endosomal pH, which can affect BACE1 activity, one of the two enzymes required to process APP to release the Aβ peptide. *Prasad and Rao, 2015*, reported that overexpression of NHE6 and full-length APP in HEK293 cells, rather than its inhibition or knockdown, reduced Aβ generation, which conflicts with our findings in primary cortical neurons (*Figure 6*). Although the cause of this discrepancy remains currently unresolved, it is possible that it is the result of the two fundamentally different experimental systems that were used in the respective studies, that is, overexpression in immortalized kidney cells on one hand and primary cortical neuronal cultures on the other. Using a humanized *App^{NL-F}* knockin mouse model, we show that NHE6/*Slc9a6* deficiency in 1-year-old animals reduces plaque deposition by ~80 %. In *App^{NL-F}* mice, plaques can be identified as early as 9 months of age. Whereas plaque deposition only increases by less than twofold between 9 and 12 months of age, it increases tenfold between 12 and 18 months (*Saito et al., 2014*). Importantly, the reduction in plaque deposition by *Slc9a6* deficiency persists from early (12 months) to later stages (18 months) of AD, as *Slc9a6*-deficient *App^{NL-F}* animals aged 18 months had a reduction in insoluble Aβ by approximately 71 %. NHE6/*Slc9a6* depletion in *App^{NL-F}*;*Apoe^{APOE4}* mice showed a comparable reduction in plaque load (*Figure 7* and *Figure 7—figure supplement 3*). Our data are consistent with previous findings by the Holtzman group (*Fagan et al., 2002*) that showed that mouse ApoE promotes plaque deposition more potently than human ApoE4.

Prevention of plaque formation in our *Slc9a6*-deficient model was likely caused by increased microglial activation and plaque phagocytosis (*Figure 7*, *Figure 7—figure supplement 3*, *Figure 8*, *Figure 9*, *Figure 9—figure supplement 1*, and *Figure 9—figure supplement 2*). In the brains of AD patients and APP-overexpressing mice, plaques are surrounded by reactive microglia and astrocytes (*Meyer-Luehmann et al., 2008*; *Serrano-Pozo et al., 2013*), but the pathological significance of this is incompletely understood. Beneficial or detrimental roles of reactive microglia and astrocytes in

the degradation of Aβ have been reported, depending on the activation state of these cells (*Meyer-Luehmann et al., 2008*; *Ziegler-Waldkirch and Meyer-Luehmann, 2018*). We observed an increase in reactive microglia and astrocytes resulting from NHE6/*Slc9a6* depletion, which correlated with reduced plaque deposition in *App^NL-F* mice, irrespective of the presence of either murine ApoE or human ApoE4. As murine ApoE exacerbates plaque deposition even more than ApoE4, the comparable plaque reduction in *Slc9a6*-deficient mice with murine *Apoe* or *Apoe^APOE4* might be the result of a maximally accelerated early endosomal maturation and cargo transport in the absence of NHE6. When compared to control *App^NL-F* mice, *Slc9a6*-deficient *App^NL-F* mice show an increase in Iba1 (microglia) and GFAP (astrocytes) immunoreactivity, but reduced Aβ immunoreactivity. Surprisingly, the intensity of GFAP and Iba1 in plaque areas was comparable between the groups. Moreover, even though Aβ is reduced and Iba1 is increased in the *Slc9a6^-*, the proportion of microglial structures containing Aβ (6E10 antibody) was comparable between *Slc9a6*-deficient and control *App^NL-F* mice, as was the intensity signal for 6E10. This suggests that microglia in the *Slc9a6^-* may be more efficient in taking up and degrading Aβ. Whether the reduction in plaques is due to the presence of an increased number of microglia and astrocytes that actively phagocytose nascent plaques, or whether endosomal acidification in microglia and astrocytes improves their ability to degrade or export Aβ from the brain remains to be determined. It is tempting to speculate that the mechanism that leads to reduced plaque load in *Slc9a6*-deficient *App^NL-F* mice may involve an increased catabolic rate (*Shi et al., 2021*), brought about by the accelerated acidification and vesicular trafficking of EEs.

It is also possible that *Slc9a6* deficiency alters the efficiency of astrocytes to lipidate ApoE. In a mouse model, improved ApoE lipidation by the overexpression of ATP-binding cassette transporter family member A1 (ABCA1) decreased plaque deposition (*Wahrle et al., 2005*; *Wahrle et al., 2008*; *Wahrle et al., 2004*). During HDL assembly, ABCA1 shuttles between EE and the plasma membrane, a process also referred to as retroendocytosis (*Ouimet et al., 2019*). Moreover, membrane trafficking of ABCA1 is altered by ApoE in an isoform-dependent fashion (*Rawat et al., 2019*). The *App^NL-F* mouse model used in our study develops plaques at 12 months (*Saito et al., 2014*) in the presence of murine ApoE. However, the onset of plaque deposition in human *Apoe^APOE4* mice was delayed by ~3 months. The effect of germline *Slc9a6* deficiency and conditional *Slc9a6* deficiency induced at 2 months had a comparable effect on plaque reduction in *App^NL-F* and *App^NL-F*;*Apoe^APOE4* mice.

*Slc9a6^-* and *Slc9a6^fl*;*CAG-Cre^ERT2* both show progressive Purkinje cell loss in the cerebellum, indicating that NHE6 requirement is cell-autonomous and not developmentally determined. *Slc9a6^-* and *Slc9a6^fl*;*CAG-Cre^ERT2* show a comparable increase in immunoreactivity against Iba1 and GFAP. Increased glia reactivity can be a direct cell-autonomous effect of NHE6 loss or an indirect effect caused by NHE6 deficiency-related neurodegeneration. As germline *Slc9a6^-* mice present with a reduction in cortical thickness and hippocampal volume caused by both neurodevelopmental and neurodegenerative effects (*Xu et al., 2017*; *Figure 7—figure supplement 1*), it is possible that neurodegeneration gives rise to glia activation. However, the tamoxifen-induced *Slc9a6^fl*;*CAG-Cre^ERT2* mice do not present with a reduction in cortical or hippocampal volume (*Figure 7*), yet have comparable immunoreactivity for markers of glial activation. Our study supports a temporally distinct function of NHE6 in the adult brain where the cerebrum requires NHE6 for development, but not for postnatal neuronal survival, whereas Purkinje cells in the cerebellum do. A similar dual function for NHE6 has been described previously by *Xu et al., 2017*. Moreover, this further indicates that reduced plaque load is also not an effect of neuronal loss. Therefore, our two mouse models together suggest that the observed increased glial activation is not caused by neuronal cell loss, but rather is likely a direct cell-autonomous effect of NHE6 loss of function. It thus remains to be determined whether endosomal acidification in NHE6-deficient microglia alone is sufficient to induce Aβ degradation and plaque reduction.

NHE6 is ubiquitously expressed in all cells of the body, however, in the CNS it is highly expressed in neurons where abundant synaptic vesicles and neurotransmitter receptors recycle in synapses (*Lee et al., 2020*; *Lee et al., 2021*) and to a lesser extent in glial cells (*Zhang et al., 2014*). Neuronal NHE6 plays a direct role in both synaptic development and plasticity, potentially through BDNF/TrkB (*Deane et al., 2013*; *Ouyang et al., 2013*) and other pathways. The impact of NHE6 loss in microglia and astrocytes is still unknown. Global *Slc9a6* deficiency causes glial activation in vivo, which could be mediated through either a primary cell-autonomous mechanism or a secondary mechanism induced by damaged neurons. In ApoE4-expressing astrocytes, others have shown that overexpression of

NHE6 increases LRP1 on the surface, which correlated with an increase in Aβ uptake (**Prasad and Rao, 2018**). Microglia are the primary glial cells that degrade Aβ; thus, it will be imperative to determine how NHE6 levels alter Aβ degradation selectively in astrocytes and microglia, respectively.

Early endosomal pH balance in phagocytic cells plays an important role in viral and bacterial infection response. In phagocytic cells, the deacidification of endosomes using pharmacological inhibitors like chloroquine has been shown to reduce endosomal toll-like receptor (TLR) response (**de Bouteiller et al., 2005**; **Fox, 2019**; **Kuznik et al., 2011**; **Wozniacka et al., 2006**; **Yang et al., 2016**), suggesting that NHE6 depletion in microglia might conversely augment TLR response.

It would also be intriguing to test whether *Slc9a6* deficiency increases cognitive performance in AD mouse models. In the current study we used $App^{NL-F}$ and $App^{NL-F};Apoe^{APOE4}$ mice as the most physiological currently available mouse models of AD that do not rely on excessive amyloid overproduction. These mice, however, do not show cognitive impairments in spatial learning tests like Morris water maze (**Saito et al., 2014**) (own unpublished observations). Other neurobehavioral phenotypes have been described for $Slc9a6^{-}$ mice, which also recapitulate symptoms in Christianson syndrome patients, for example, hyposensitivity to pain (**Petitjean et al., 2020**). Future studies on our novel tamoxifen-inducible $Slc9a6^{fl};CAG-Cre^{ERT2}$ line will help to understand whether these symptoms are based on neurodevelopmental deficits caused by germline *Slc9a6* deficiency or whether they can be reproduced by induced loss of NHE6 postnatally.

In conclusion, we have shown that both the endosomal trafficking defect induced by ApoE4 in neurons and increased plaque deposition irrespective of ApoE genotype can be corrected by inhibition or genetic deletion of NHE6/*Slc9a6*, a key regulator of early endosomal pH. Accelerated acidification of EEs abolishes the ApoE4-induced Reelin resistance and restores normal synaptic plasticity in ApoE4-targeted replacement mice. The first FDA-approved drug for AD treatment in 18 years is aducanumab (**Sevigny et al., 2016**), an antibody directed against Aβ, which clears amyloid from the brain. However, amyloid removal in individuals already afflicted with AD provides at best marginal benefits at this stage. Moreover, in excess of 1 billion people worldwide are ApoE4 carriers, making early treatment with a complex biologic such as aducanumab impractical on the global scale. Here, we have presented a potential alternative approach which should be adaptable to large-scale prevention treatment using blood-brain barrier penetrant NHE6-specific inhibitors. Taken together, our combined data suggest that endosomal acidification has considerable potential as a novel therapeutic approach for AD prevention and possibly also for the prevention of disease progression.

## Materials and methods

**Key resources table**

| Reagent type (species) or resource | Designation | Source or reference | Identifiers | Additional information |
|---|---|---|---|---|
| Strain, strain background (*Mus musculus*) | Mouse/$Slc9a6^{fl}$ | This study | | Refer to Materials and methods section for detailed description of mouse model production |
| Strain, strain background (*Mus musculus*) | Mouse/$Slc9a6^{-}$ | This study | | Refer to Materials and methods section for detailed description of mouse model production |
| Strain, strain background (*Mus musculus*) | Mouse/$Apoe^{APOE3}$ | **Sullivan et al., 1997** | IMSR_TAC:2,542 | $Apoe^{APOE3}$ |
| Strain, strain background (*Mus musculus*) | Mouse/$Apoe^{APOE4}$ | **Knouff et al., 1999** | IMSR_TAC:3,518 | $Apoe^{APOE4}$ |
| Strain, strain background (*Mus musculus*) | Mouse/B6.Cg-Gt(ROSA)26Sortm9(CAG-tdTomato)Hze/J | The Jackson Laboratory **Madisen et al., 2010** | JAX #007909 | $ROSA^{floxedStop-tdTomato}$ |
| Strain, strain background (*Mus musculus*) | Mouse/CAG-cre/Esr15Amc/J | The Jackson Laboratory **Hayashi and McMahon, 2002** | JAX #004682 | $CAG-Cre^{ERT2}$ |

*Continued on next page*

*Continued*

| Reagent type (species) or resource | Designation | Source or reference | Identifiers | Additional information |
|---|---|---|---|---|
| Strain, strain background (*Mus musculus*) | Mouse/B6.129S4-Meox2tm1(cre)Sor/J | The Jackson Laboratory **Tallquist and Soriano, 2000** | JAX 003755 | *Meox-Cre* |
| Strain, strain background (*Mus musculus*) | *App^{NL-F}* | **Saito et al., 2014** | | *App^{NL-F}* |
| Strain, strain background (*Mus musculus*) | Tg2576 | Charles River **Hsiao et al., 1996** | Charles River Tg2576 | Tg2576, *APPSwe* |
| Strain, strain background (*Rattus norvegicus*) | SD rat | Charles River | SC:400 | |
| Cell line (*Homo sapiens*) | HEK293 | Thermo Fisher | R70507, RRID:CVCL_0045 | |
| Cell line (*Homo sapiens*) | HEK293-T | ATCC | CRL-3216 | |
| Cell line (*Mus musculus*) | Neuro-2a | ATCC | CCL-131 | |
| Cell line (*Mus musculus*) | NHE6-KO (*Slc9a6^-*) mouse embryonic fibroblasts (MEFs) | This study | | Refer to Materials and methods section for detailed description of MEF production |
| Cell line (*Mus musculus*) | *Slc9a6^+* MEFs (*Slc9a6^-* littermate) | This study | | Refer to Materials and methods section for detailed description of MEF production |
| Antibody | Anti-Aβ (clone 6E10) (mouse monoclonal) | Covance | SIG-39320 RRID:AB_662798 | WB and IHC (1:1000) |
| Antibody | Anti-Aβ (clone 4 G8) (mouse monoclonal) | Covance | SIG-39220 RRID:AB_10175152 | IHC (1:1000) |
| Antibody | Anti-phospho tyrosine (clone 4 G10) (mouse monoclonal) | EMD Millipore | Millipore Cat# 05–321, RRID:AB_309678 | WB (1:1000) |
| Antibody | Anti-Apoer2 (rabbit polyclonal) | Herz Lab, #2561, **Trommsdorff et al., 1999** | | WB (1:1000) |
| Antibody | Anti-β-Actin (rabbit polyclonal) | Abcam | Ab8227, RRID:AB_2305186 | WB (1:3000) |
| Antibody | Anti-Calbindin D-28k (mouse monoclonal) | Swant | Swant Cat# 300, RRID:AB_10000347 | IHC (1:1000) |
| Antibody | Anti-GFAP (rabbit polyclonal) | Abcam | Abcam Cat# ab7260, RRID:AB_305808 | IHC (1:2000) |
| Antibody | Anti-GluA1 (rabbit polyclonal) | Abcam | ab31232, RRID:AB_2113447 | WB (1:1000) |
| Antibody | Anti-GluA2/3 (rabbit polyclonal) | EMD Millipore | 07–598, RRID:AB_310741 | WB (1:1000) |
| Antibody | Anti-GluN2B (rabbit polyclonal) | Cell Signaling Technology | 4,207 S, RRID:AB_1264223 | WB (1:1000) |
| Antibody | Anti-Iba1 (rabbit polyclonal) | Wako | 019–19741, RRID:AB_839504 | IHC (1:1000) |
| Antibody | Anti-NHE6 (C-terminus) (rabbit polyclonal) | Herz Lab, **Xian et al., 2018** | | WB (1:1000) |
| Antibody | Anti-mouse-IgG AF594 (goat polyclonal) | Thermo Fisher | A-11032, RRID:AB_2534091 | IHC (1:500) |

*Continued*

| Reagent type (species) or resource | Designation | Source or reference | Identifiers | Additional information |
|---|---|---|---|---|
| Antibody | Anti-rabbit-IgG AF488 (goat polyclonal) | Thermo Fisher | A-11034, RRID:AB_2576217 | IHC (1:500) |
| Commercial assay or kit | Anti-mouse-IgG staining kit | Vector | MP-7602, RRID:AB_2336532 | |
| Commercial assay or kit | Anti-rabbit-IgG staining kit | Vector | MP-7601, RRID:AB_2336533 | |
| Chemical compound, drug | Antigen retrieval citrate buffer | BioGenex, Cat | HK086-9K | |
| Chemical compound, drug | B-27 Supplement (50 ×), serum free | Thermo Fisher | 17504044 | |
| Chemical compound, drug | Cytoseal 60 | Thermo Fisher | 8310 | |
| Chemical compound, drug | DMEM | Sigma-Aldrich | D6046 | |
| Chemical compound, drug | FuGENE | Promega | E2311 | |
| Chemical compound, drug | HBSS (1 ×) | Gibco | 14175 | |
| Chemical compound, drug | L-Glutamic acid (glutamate) | Sigma-Aldrich | G1251 | |
| Chemical compound, drug | γ-Secretase inhibitor L-685458 | Tocris Bioscience | 2627 | |
| Chemical compound, drug | Penicillin-streptomycin solution, 100 × | Corning | 30–002 CI | |
| Chemical compound, drug | Neurobasal Medium (1 ×) liquid without Phenol Red | Thermo Fisher | 12348017 | |
| Chemical compound, drug | NeutrAvidin Agarose | Thermo Fisher | 29201 | |
| Chemical compound, drug | Nonidet P-40 Alternative | EMD Millipore | 492016 | |
| Chemical compound, drug | 32 % Paraformaldehyde AQ solution | Fisher Scientific | 15714 S | |
| Chemical compound, drug | PBS (1 ×) | Sigma-Aldrich | D8537 | |
| Chemical compound, drug | Penisillin-streptomycin | Corning | 30–002 CI | |
| Chemical compound, drug | Phosphatase inhibitor cocktail | Thermo Fisher | 78420 | |
| Chemical compound, drug | Poly-D-lysine | Sigma-Aldrich | A-003-M | |
| Chemical compound, drug | Protein A-Sepharose 4B | Thermo Fisher | 101042 | |
| Chemical compound, drug | Proteinase Inhibitor Cocktail | Sigma-Aldrich | P8340 | |
| Chemical compound, drug | Sulfo-NHS-SS-biotin | Pierce | 21331 | |
| Chemical compound, drug | Triton X-100 | Sigma-Aldrich | CAS9002-93-1 | |

*Continued on next page*

*Continued*

| Reagent type (species) or resource | Designation | Source or reference | Identifiers | Additional information |
|---|---|---|---|---|
| Chemical compound, drug | Tween 20 | Sigma | P1379 | |
| Other | Vectashield with DAPI | Vector Labs | H-1200 | (DAPI 1.5 µg/ml) |
| Transfected construct (*Mus musculus*) | pCrl, Reelin expression vector | *D'Arcangelo et al., 1997* | N/A | |
| Transfected construct (*Homo sapiens*) | pcDNA3.1-ApoE3 | *Chen et al., 2010* | N/A | Progenitor pcDNA3.1-Zeo |
| Transfected construct (*Homo sapiens*) | pcDNA3.1-ApoE4 | *Chen et al., 2010* | N/A | Progenitor pcDNA3.1-Zeo |
| Transfected construct (*Mus musculus*) | pLKO.1 scramble shRNA | *Xian et al., 2018* | N/A | |
| Transfected construct (*Mus musculus*) | pLKO.1 shNHE6 | Open Biosystem *Xian et al., 2018* | TRCN0000068828 | Refer to shNHE6-a |
| Transfected construct (*Mus musculus*) | psPAX2 | Addgene | 12260 | Plasmid was a gift from Didier Trono |
| Transfected construct (*Mus musculus*) | pMD2.G | Addgene | 12259 | Plasmid was a gift from Didier Trono |
| Transfected construct (*Mus musculus*) | pJB-NHE6 targeting vector | This study | N/A | Refer to Materials and methods section for detailed description |
| Recombinant DNA reagent | pJB1 (plasmid) | *Braybrooke et al., 2000* | N/A | |
| Recombinant DNA reagent | pCR4-TOPO (plasmid) | Thermo Fisher | K457502 | |
| Recombinant DNA reagent | pLVCMVfull (plasmid) | *Xian et al., 2018* | N/A | |
| Recombinant DNA reagent | pME (plasmid) | *Stawicki et al., 2014* | Addgene #73794 | Plasmid was a gift from David Raible |
| Recombinant DNA reagent | pLVCMV Vamp3pHluorin2 (plasmid) | This study | N/A | Refer to Materials and methods section for detailed description |
| Recombinant DNA reagent | BAC containing murine NHE6 sequence (bacterial artificial chromosome) | BACPAC Resources Center | RP23 364 F14 | |
| Software, algorithm | Adobe Creative Cloud | Adobe | RRID:SCR_010279 | |
| Software, algorithm | GraphPad Prism 7.0 | GraphPad Software | RRID:SCR_002798 | |
| Software, algorithm | Fiji/ImageJ | NIH | RRID:SCR_002285 | |
| Software, algorithm | LabView7.0 | National Instruments | RRID:SCR_014325 | |
| Software, algorithm | NDP.view2 | Hamamatsu Photonics | | |
| Software, algorithm | Odyssey Imaging System | LI-COR | RRID:SCR_014579 | |
| Software, algorithm | Clustal Omega | EMBL-EBI | RRID:SCR_001591 | |

*Continued on next page*

*Continued*

| Reagent type (species) or resource | Designation | Source or reference | Identifiers | Additional information |
|---|---|---|---|---|
| Software, algorithm | Leica TCS SPE | Leica | RRID:SCR_002140 | |
| Sequence-based reagent | SA forward | IDT | | GGATCCGTGT GTGTGTTGGG GGAGGGA |
| Sequence-based reagent | SA reverse | Integrated DNA Technology | | CTCGAGCTCAC AATCAGCCCTTT AAATATGCC |
| Sequence-based reagent | GAP repair US forward | Integrated DNA Technology | | AAGCTTGCGGCC GCTTCAATTTCTG TCCTTGCTACTG |
| Sequence-based reagent | GAP repair US reverse | Integrated DNA Technology | | AGATCTCAAGAA AGTTAGCTAGA AGTGTGTC |
| Sequence-based reagent | GAP repair DS forward | Integrated DNA Technology | | AGATCTGTAGA GGATGTGGGA AAGAGAG |
| Sequence-based reagent | GAP repair DS reverse | Integrated DNA Technology | | GTCGACGCGG CCGACACACA CAGATAAATAA CCTCAAAAG |
| Sequence-based reagent | 5' flanking 1st LoxP fragment forward | Integrated DNA Technology | | GCTTCTCTCG AGCAAGAGTCAAC |
| Sequence-based reagent | 5' flanking 1st LoxP fragment reverse | Integrated DNA Technology | | GATATCAGCA GGTACCACCAA GATCTCAACCT TATTGTCCTATA TGCACAAAC |
| Sequence-based reagent | 3' flanking 1st LoxP fragment forward | Integrated DNA Technology | | GTCTTGTTGGTA CCTGATGAAATG GACTACCTCCACTTG |
| Sequence-based reagent | 3' flanking 1st LoxP fragment reverse | Integrated DNA Technology | | ATCGATCTTCA TAACCCATCTGGATA |
| Sequence-based reagent | LoxP Oligo forward | Integrated DNA Technology | | GATCTGCTCAGC ATAACTTCGTATAG CATACATTATACG AAGTTATGGTAC |
| Sequence-based reagent | LoxP Oligo reverse | Integrated DNA Technology | | CATAACTTCGTA TAATGTATGCTAT ACGAAGTTATGC TGAGCAGATC |
| Sequence-based reagent | Genotyping NHE6-floxed and wt forward | Integrated DNA Technology | | GAGGAAGC AAAGTGTCA GCTCC |
| Sequence-based reagent | Genotyping NHE6-floxed and wt reverse | Integrated DNA Technology | | CTAATCCCCTC GGATGCTGCTC |
| Sequence-based reagent | Genotyping NHE6-KO forward | Integrated DNA Technology | | GAGGAAGC AAAGTGTCA GCTCC |
| Sequence-based reagent | Genotyping NHE6-KO reverse | Integrated DNA Technology | | CCTCACAAGACT AGAGAAATGGTTC |

*Continued on next page*

*Continued*

| Reagent type (species) or resource | Designation | Source or reference | Identifiers | Additional information |
|---|---|---|---|---|
| Sequence-based reagent | Vamp3 forward | Integrated DNA Technology | | TTCAAGCTTCAC CATGTCTACAGG TGTGCCTTCGGGGTC |
| Sequence-based reagent | Vamp3 reverse | Integrated DNA Technology | | CATTGTCATCAT CATCATCGTGTG GTGTGTCTCTAA GCTGAGCAACAG CGCCGTGGACGGC ACCGCCGGCCCCG GCAGCATCGCCAC CAAGCTTAAC |
| Sequence-based reagent | pHluorin2 forward | Integrated DNA Technology | | CCGGTCCCAAGCTT ATGGTGAGCAAGG GCGAGGAGCTGTTC |
| Sequence-based reagent | pHluorin2 reverse | Integrated DNA Technology | | GCCCTCTTCTAGAG AATTCACTTGTACAG CTCGTCCATGCCGTG |

## Animals

All animal procedures were performed according to the approved guidelines (Animal Welfare Assurance Number D16-00296) for Institutional Animal Care and Use Committee (IACUC) at the University of Texas Southwestern Medical Center at Dallas.

The mouse lines Rosa-stop-tdTomato *B6.Cg-Gt(ROSA)26Sor^{tm9(CAG-tdTomato)Hze}/J* (*Madisen et al., 2010*) (JAX #007909) and CAG-Cre^{ERT2} *B6.Cg-Tg(CAG-cre/Esr1)5Amc/J* mice (*Hayashi and McMahon, 2002*) (JAX #004682) were obtained from The Jackson Laboratories (Bar Harbor, ME). ApoE3 and ApoE4 targeted replacement mice (*Apoe^{APOE3}*, *Apoe^{APOE4}*) (*Knouff et al., 1999*; *Sullivan et al., 1997*) were kind gifts of Dr Nobuyo Maeda. *AppSwe* (*Tg2576*) were generated by *Hsiao et al., 1996*. The Meox-Cre *B6.129S4-Meox2tm1(cre)Sor/J* mice (JAX 003755) were provided by Drs M Tallquist and P Soriano (*Tallquist and Soriano, 2000*). Conditional NHE6 KO (*Slc9a6^{fl}*;*CAG-Cre^{ERT2}*) and germline NHE6 KO (*Slc9a6^-*) mice were generated as described below. The human APP knockin line (*App^{NL-F}*) (*Saito et al., 2014*) has been described earlier. Pregnant female SD (Sprague Dawley) rats were obtained from Charles River (SC:400). Mice were group-housed in a standard 12 hr light/dark cycle and fed ad libitum standard mouse chow (Envigo, Teklad 2016 diet as standard and Teklad 2018 diet for breeding cages).

To generate *Slc9a6*-floxed (*Slc9a6^{fl}*) mice, the first exon of NHE6 was flanked with loxP sites (*Figure 2A*). A loxP site was inserted 2 kb upstream of the first exon of the X-chromosomal NHE6 gene and a Neo-cassette (flanked with loxP and FRT sites) was inserted 1 kb downstream of the first exon. Insertion sites were chosen based on low conservation (mVISTA) between mammalian species (rat, human, mouse). To create the targeting vector for the *Slc9a6^{fl}* mouse line, pJB1 (*Braybrooke et al., 2000*) was used as backbone. Murine C57Bl/6 J embryonic stem (ES) cell DNA was used as template to amplify the short arm of homology (SA; 0.88 kb of the first intron starting 1 kb 3' downstream of exon 1), which was inserted between the Neo and HSVTK selection marker genes of pJB1 (BamHI and XhoI sites) to create an intermediate plasmid referred to as pJB1-NHE6SA. To create the intermediate plasmid pNHE6-LA for the long homology arm, a fragment spanning from 13 kb 5' upstream to 1 kb 3' downstream of the first *Slc9a6* (NHE6) exon was integrated into pCR4-TOPO (Thermo Fisher) by using a bacterial artificial chromosome (RP23 364F14, Children's Hospital Oakland Research Institute [CHORI]) and the GAP repair technique (*Lee et al., 2001*) (primers to amplify the upstream (US) and downstream (DS) homology boxes are listed in the Key resources table). In a parallel cloning step, a 2.4 kb XhoI-EcoRV *Slc9a6* promoter region fragment spanning from 2.7 to 0.4 kp 5' upstream of the *Slc9a6* start-codon was modified with the 1st loxP site to generate pLoxP: three fragments (1) 0.7 kb 5' loxP flanking *Slc9a6*-fragment, (2) 1.7 kb 3' loxP flanking *Slc9a6* fragment, and (3) 100 bp loxP oligo (primers/oligos for each fragment are listed in the Key resources table) were cloned into pCR4-TOPO. The 2.5 kb loxP-modified XhoI-EcoRV *Slc9a6*-promoter fragment of pLoxP was cloned into pNHE6-LA to create pNHE6-LA-LoxP. To obtain

the final targeting construct pJB-NHE6-TV, the NotI-EagI fragment of pNHE6-LA-LoxP containing the long arm (11 kb 5′ upstream of the 1st LoxP) and the 1st loxP site was cloned into the NotI-site of pJB1-NHE6SA and checked for orientation (pJB1-NHE6 targeting vector). pJB-NHE6-TV was linearized with NotI and electroporated into C57Bl/6 J ES cells. Gene targeting-positive C57Bl/6 J ES cells (PCR screen) were injected into albino C57Bl/6 J blastocysts, resulting in chimeric mice. The chimeras were crossed to C57Bl/6 J mice, resulting in *Slc9a6*$^{fl/+}$ females. Genotyping: The *Slc9a6*$^{fl}$ PCR amplifies a 250 bp of the wildtype and 270 bp of the floxed allele, primers are listed in the Key resources table.

To generate *Slc9a6*$^{fl}$;*CAG-Cre*$^{ERT2}$, *Slc9a6*$^{fl/+}$ females were crossed to *CAG-Cre*$^{ERT2}$ mice to obtain *Slc9a6*$^{fl/fl}$;*CAG-Cre*$^{ERT2}$ and *Slc9a6*$^{y/fl}$;*CAG-Cre*$^{ERT2}$ mice (*Slc9a6*$^{fl}$;*CAG-Cre*$^{ERT2}$) and *CAG-Cre*$^{ERT2}$ negative control littermates (*Slc9a6*$^{fl}$). *Slc9a6*$^{fl}$;*CAG-Cre*$^{ERT2}$ mice were backcrossed to *Apoe*$^{APOE3}$ or *Apoe*$^{APOE4}$ mice. Breeding pairs were set up in which only one parent was *CAG-Cre*$^{ERT2}$ positive. The following genotypes were used for hippocampal field recordings: (1) *Apoe*$^{APOE3}$;*SLC9a6*$^{y/fl}$ (short:*Apoe*$^{APOE3}$), (2) *Apoe*$^{APOE3}$;*Slc9a6*$^{y/fl}$;*Cre*$^{ERT2}$ (short: *Apoe*$^{APOE3}$; *Slc9a6*$^{fl}$;*CAG-Cre*$^{ERT2}$), (3) *Apoe*$^{APOE4}$;*Slc9a6*$^{y/fl}$ (short: *Apoe*$^{APOE4}$), and (4) *Apoe*$^{APOE4}$;*SLc9a6*$^{y/fl}$;*Cre*$^{ERT2}$ (short: *Apoe*$^{APOE4}$;*Slc9a6*$^{fl}$;*CAG-Cre*$^{ERT2}$). The *Apoe*$^{APOE4}$;*Slc9a6*$^{fl}$;*CAG-Cre*$^{ERT2}$ line was further crossed with the *App*$^{NL-F}$ line to generate *Apoe*$^{APOE4}$;*Slc9a6*$^{fl}$;*CAG-Cre*$^{ERT2}$;*App*$^{NL-F}$ and *Apoe*$^{APOE4}$;*Slc9a6*$^{fl}$;*App*$^{NL-F}$ control littermates. To induce genetic depletion of NHE6, tamoxifen (120 mg/kg) was intraperitoneally injected at 6–8 weeks of age. Injections were applied for 5 consecutive days. Tamoxifen was dissolved in sunflower oil (Sigma, W530285) and 10 % EtOH.

To generate the germline NHE6 KO line (*Slc9a6*$^{-}$), heterozygous *Slc9a6*$^{fl/+}$ females were crossed to *Meox-Cre* to yield germline mutant *Slc9a6*$^{-/-}$ females and *Slc9a6*$^{y/-}$ males (*Slc9a6*$^{-}$). Genotyping: The *Slc9a6*$^{fl}$ PCR amplifies 250 bp of the wildtype, 270 bp of the floxed, and no fragment in the KO alleles. Recombination was verified with the NHE6-rec PCR, which amplifies 400 bp if recombination has occurred. PCR primers are listed in the Key resources table. *Slc9a6*$^{-}$ animals were further crossed with *App*$^{NL-F}$ (**Saito et al., 2014**) mice. *App*$^{NL-F}$;*Slc9a6*$^{-}$ (*App*$^{NL-F/NL-F}$;*Slc9a6*$^{y/-}$ or *App*$^{NL-F/NL-F}$;*Slc9a6*$^{-/-}$) and control (*App*$^{NL-F}$ = *AppNL*$^{NL-F/NL-F}$;*Slc9a6*$^{y/+}$ or *App*$^{NL-F/NL-F}$;*Slc9a6*$^{+/+}$) littermates were obtained by crossing *Slc9a6*$^{+/-}$;*App*$^{NL-F/NL-F}$ females with either *Slc9a6*$^{y/+}$;*App*$^{NL-F/NL-F}$ or *Slc9a6*$^{y/-}$;*App*$^{NL-F/NL-F}$ males.

## DNA constructs, recombinant proteins, lentivirus production

Lentiviral plasmids with shRNA directed against NHE6/*Slc9a6* and the scrambled control have been described in **Xian et al., 2018**. Plasmids encoding ApoE3 and ApoE4 (**Chen et al., 2010**), and Reelin **D'Arcangelo et al., 1997** have been described before. The lentiviral plasmid encoding the Vamp3-pHluorin2 fusion protein was cloned by inserting mouse Vamp3 cDNA, a linker and pHluorin2 (pME, Addgene #73794) (**Stawicki et al., 2014**) into pLVCMVfull (**Xian et al., 2018**). For Vamp3 the forward primer (5′- TTCAAGCTTCACCATGTCTACAGGTGTGCCTTCGGGGTC-3′) contains a Kozak sequence, the reverse primer encodes a KLSNSAVDGTAGPGSIAT linker (**Nakamura et al., 2005**) (5′ CATTGTCATCATCATCATCGTGTGGTGTGTCTCTAAGCTGAGCAACAGCGCCGTGGACGG CACCGCCGGCCCCGGCAGCATCGCCACCAAGCTTAAC-3') . The pHluorin2 primers were forward 5′-CCGGTCCCAAGCTTATGGTGAGCAAGGGCGAGGAGCTGTTC-3′ and reverse 5′- GCCCTCTT CTAGAGAATTCACTTGTACAGCTCGTCCATGCCGTG-3′. The fragments were sequentially cloned into pcDNA3.1 and the fusion protein was then transferred with NheI and EcoRI into pLVCMV-full. Lentiviral plasmids psPAX2 and pMD2.g were a kind gift from Dr D Trono and obtained from Addgene.

Recombinant Reelin and ApoE were generated in HEK293 cells. Reelin was purified as described before (**Weeber et al., 2002**). ApoE-conditioned medium was collected from HEK293 cell cultures transiently transfected with pcDNA3.1-ApoE constructs or empty control vector (pcDNA3.1-Zeo). ApoE concentration was measured as described before (**Xian et al., 2018**).

For lentivirus production HEK 293 T cells were co-transfected with psPAX2, pMD2.g, and the individual shRNA encoding transfer or Vamp3-pHluorin2 constructs. Media was replaced after 12–16 hr. Viral particle containing media was collected after centrifuging cellular debris. The viral particles were 10 × concentrated by ultra-centrifugation and resuspended in Neurobasal medium. To infect neurons on DIV7 1/10th of the culture medium was replaced by concentrated virus. After 24 hr, the virus was removed.

## Histochemistry

Mice were euthanized with isoflurane and perfused with PBS followed by 4 % paraformaldehyde (PFA) in PBS. Brains were removed and post-fixed for 24 hr in 4 % PFA. Fresh fixed brains were immobilized in 5 % agarose in PBS and 50 µm thick sections were sliced on a vibratome (Leica, Wetzlar, Germany). Vibratome slices of Rosa26$^{floxedStop-tdTomato}$;CAGCre$^{ERT2}$ mice, with or without tamoxifen injection at 8 weeks of age were mounted with Antifade Mounting medium containing DAPI (Vectashield). For immunofluorescence, vibratome slices were labeled for Calbindin after permeabilization with 0.3 % Triton X in PBS and blocking for 1 hr in blocking buffer (10 % normal goat serum, 3 % BSA, and 0.3 % Triton X in PBS). The primary antibody mouse anti-Calbindin (Swant CB300) was diluted in blocking buffer (1:1000) and added to the slices for 24–48 hr at 4 °C. Slices were subsequently washed 4 × 15 min with PBS containing 0.3 % Triton X. Slices were incubated with anti-mouse IgG coupled to Alexa594 (1:500 in blocking buffer) for 2 hr at room temperature. After washing, slices were mounted with Antifade Mounting medium with DAPI (Vectashield). Images were taken with an Axioplan two microscope (Zeiss).

For immunohistochemistry staining, PFA-fixed brains were block-sectioned into coronal slabs, paraffin-embedded, and serially sectioned on a rotating microtome (Leica) at 5 µm. Deparaffinized sections were stained with Thioflavin S as described before (*Guntern et al., 1992*). Briefly, deparaffinized slices were oxidized in 0.25 % $KMnO_4$ for 20 min. After washing with water, slices were bleached with 1 % $K_2S_2O_5$ / $C_2H_2O_4$ for 2 min, washed in water, and treated with 2 % $NaOH/H_2O_2$ for 20 min. After washing with water, slices were acidified in 0.25 % $CH_3COOH$ for 1 min, washed with water and equilibrated in 50 % EtOH for 2 min, and stained in Thioflavin S solution for 7 min. Reaction was stopped by washing in 50 % EtOH. Slices were dehydrated with 95 % EtOH, followed by 100 % EtOH and xylene. Slices were mounted with Cytoseal 60 (Thermo Scientific). Deparaffinized sections were labeled using antibodies raised against GFAP (Abcam AB5804, Rabbit, 1:2000), Iba1 (Wako 019–19741, Rabbit, 1:1000), and Aβ (4G8, Covance or 6E10, Biolegend, both mouse, 1:1000). Briefly, 5 µm sections were deparaffinized, subjected to microwave antigen retrieval (citrate buffer, pH 6.0), permeabilized with 0.3 % (vol/vol) Triton X, endogenous peroxidases activity was quenched for diaminobenzidine (DAB) staining. Slices were blocked with goat serum (2.5%) prior to overnight incubation with primary antibodies at 4 °C. Primary antibodies were detected by either fluorescent secondary antibodies (goat-anti-mouse Alexa594, goat-anti-rabbit Alexa488) or sequential incubation with biotinylated secondary antisera and streptavidin-peroxidase for DAB staining. DAB chromagen was used to detect the immunoperoxidase signal (*Sinclair et al., 1981*) (Vector; anti-mouse and anti-rabbit IgG kits). Fluorescence-labeled slices were counterstained with DAPI (mounting media with DAPI, Vectashield). Standard protocols were utilized for staining of paraffin sections with H&E (Leica) (*Fischer et al., 2008*). Microscopy was performed on a high-throughput microscope (NanoZoomer 2.0-HT, Hamamatsu) for DAB-stained tissue or with an Axioplan two microscope (Zeiss) for immunofluorescence analysis. The analysis of DAB-labeled antibodies was performed after exporting the images with NDP.view2 software with ImageJ. For Thioflavin S staining and Aβ labeling plaques were quantified by categorizing them as small, medium, and large (Thioflavin S) or dense and diffuse (4G8) as depicted in *Figure 7—figure supplement 2*. Co-localization analysis of microglia (Iba1) and astrocytes (GFAP) with plaques (6E10) was performed with ImageJ. A blind observer selected the area of plaques with circles of 20, 40, or 80 µm diameter and analyzed the intensity of 6E10 and Iba1 or GFAP by using the ImageJ plugin RGB_measure. 6E10-positive microglia (Iba1) were quantified by a blind observer by first identifying microglia structures/cells in the green channel (Iba1), and then analyzing the proportion of 6E10-positive structures and signal intensity (red channel).

## Primary cortical neuronal cultures

Primary cortical neuronal cultures were prepared from rat (SD, Charles River) or various mouse lines (*wildtype, Slc9a6⁻, AppSwe*) (E18) as described previously (*Chen et al., 2005*). Neurons were cultured in poly-D-lysine coated six-well plates (1 million/9 cm$^2$) or on coverslips (30,000 neurons/1.1 cm$^2$) in presence of Neurobasal medium supplemented with B27, glutamine, and penicillin-streptomycin at 37 °C and 5 % $CO_2$. At indicated days in vitro (DIV) primary neurons were used for experiments.

## Mouse embryonic fibroblasts

Fibroblasts were isolated from *Slc9a6*⁻ and wildtype littermate control embryos (E13.5). After removing the head and the liver, the tissue was trypsinized (0.05 % trypsin-EDTA) at 4 °C overnight, followed by 30 min at 37 °C. Suspended cells were cultivated in DMEM high glucose with 15 % FCS, 2 mM L-glutamine, and Pen/Strep. Fibroblasts were serially passaged until proliferation slowed down. Immortalization was achieved by keeping fibroblasts in culture under high confluency until they overcame their growth crisis.

## pH measurements

Mouse embryonic fibroblasts derived from either *Slc9a6*⁻ or wildtype littermate embryos were infected with Vamp3-pHluorin2 lentivirus; 24–48 hr post-infection, vesicular pH was measured on a Zeiss LSM880 Airyscan confocal microscope as described in *Ma et al., 2017*. For the standard calibration curve, cells were washed and incubated with pH standard curve buffer (125 mM KCl, 25 mM NaCl, 10 μM monensin, 25 mM HEPES for pH >7.0 or 25 mM MES for pH <7.0; pH adjusted with NaOH and HCl) and imaged in 5 % $CO_2$ at 37 °C. For vesicular pH measurements, cells were washed and imaged with pH standard curve at pH 7 without monensin. Samples were excited at 405 and 488 nm with an emission of 510 nm. For quantification, six fields of view were imaged for *Slc9a6*⁺ and four fields of view for *Slc9a6*⁻ fibroblast. Between 4 and 10 pHluorin-positive vesicles were measured per field of view which resulted in n = 32 wildtype vesicle and n = 28 *Slc9a6*⁻ vesicles. The same settings were used for every image, and images were analyzed using ImageJ software. The intensity of excitations with 405 and 488 nm was measured, individual vesicles were marked as regions of interest, and the 405/488 ratio was calculated and plotted against pH for the standard curve.

## Biochemistry

To analyze receptor recycling, cell surface biotinylation was performed. At DIV10–14, primary neurons were pre-treated for 30 min with ApoE-conditioned medium (5 μg/ml) and incubated with Reelin (2 μg/ml) for an additional 30 min (see timeline in *Figure 4A*). After treatment, cells were washed with cold PBS and incubated in PBS containing sulfo-NHS-SS-biotin for 30 min at 4 °C. Subsequently excess reagent was quenched by rinsing the neurons with cold PBS containing 100 mM glycine. Neurons were lysed in 160 μl/9 cm² lysis buffer (PBS with 0.1 % SDS, 1 % Triton X-100, and protease inhibitors) at 4 °C for 20 min. Cell debris were pelleted at 14,000 rpm for 10 min at 4 °C. The protein concentration was measured using the Bradford Protein Assay (Bio-Rad). One hundred μg of total proteins were incubated with 50 μl of NeutrAvidin agarose at 4 °C for 1 hr. Agarose beads were washed three times using washing buffer (500 mM NaCl; 15 mM Tris-HCl, pH 8.0; 0.5 % Triton X-100), biotinylated surface proteins were eluted from agarose beads by boiling in 2 × SDS sample loading buffer and loaded on an SDS-PAGE gel for Western blot analysis. GST-control and GST-RAP (50 μg/ml) pre-treatment of neurons was performed for 1 hr.

To analyze BACE1 activity, β-CTF was detected by immunoblotting. BACE1 activity was examined after pharmacological NHE inhibition or genetic *Slc9a6* knockdown in primary neurons of *AppSwe* mice. For NHE inhibition DIV10 neurons were treated with 5 μg/ml ApoE4, 3 μM EMD87580 (Merck), and/or 1 μM γ-secretase inhibitor L-685458 (Merck) for 5 hr. For knockdown of NHE6/*Slc9a6* DIV7 neurons of *AppSwe* mice were infected with lentivirus encoding shRNA against NHE6 or a scrambled control shRNA. On DIV13 neurons were treated with γ-secretase inhibitor for 12 hr. Proteins were extracted for Western blot analysis: Cells were washed three times with cold PBS and lysed for 20 min on ice in RIPA buffer (50 mM Tris-HCl, pH 8.0; 150 mM NaCl; 1 % Nonidet P-40; phosphatase and proteinase inhibitors). Cell debris were pelleted at 14,000 rpm for 10 min at 4 °C. Protein concentration was measured using the Bradford Protein Assay (Bio-Rad). After adding 4× SDS loading buffer (0.1 M Tris-HCl, pH 6.8, 2 % SDS, 5 % β-mercaptoethanol, 10 % glycerol, and 0.05 % bromophenol blue), samples were denatured at 80 °C for 10 min. Proteins were separated via SDS-PAGE and transferred to a nitrocellulose membrane for Western blotting using different antibodies listed in the Key resources table.

Brain tissue was dissected and prepared for immunoblotting as follows: After removal, brains were placed in ice-cold PBS containing proteinase inhibitors. Anatomical sectioning was performed under a microscope. The hippocampus was dissected out and transversal slices were further separated into cornu ammonis (CA) 1, CA3, and dentate gyrus. Respective pieces of the same anatomical regions of

one brain were pooled, shock-frozen in liquid nitrogen, and stored at –80 C. Frozen tissue was homogenized in 10 × volume of RIPA buffer and incubated on ice for 30 min. Cell debris were pelleted for 10 min with 14,000 rpm at 4 °C. After adding 4× SDS loading buffer, the samples were denatured at 80 °C for 10 min and used for immunoblotting.

To measure soluble and insoluble Aβ species, a sequential homogenization procedure was employed. After removal of the brains from PBS perfused mice, cortical tissue was dissected and flash-frozen. Frozen cortical tissue was homogenized in TBS supplemented with phosphatase and proteinase inhibitors at 100 mg protein/ml using a glass dounce homogenizer. Crude lysate was centrifuged at 800 × $g$ for 5 min at 4 °C. The supernatant was further centrifuged at 100,000 × $g$ for 30 min at 4 °C and collected as TBS-soluble lysate (Aβ-soluble). The TBS pellet was further homogenized in 1 % Triton-TBS containing phosphatase and proteinase inhibitors, centrifuged at 100,000 × $g$ for 30 min at 4 °C and collected as 1 % Triton-soluble lysate. The Triton-pellet was incubated with 5 M guanidine-HCl rotating at RT for 1 hr. Guanidine-soluble lysate (Aβ-insoluble) was collected after centrifugation at 21,000 × $g$ for 15 min at 4 °C. The guanidine-pellet was further solubilized in 1/20th volume with 70 % formic acid (Aβ-insoluble) and centrifuged at 21,000 × $g$ for 15 min at 4 °C. Soluble and insoluble Aβ levels were measured in duplicates using a commercial Aβ42 ELISA kit (Thermo Fisher, KHB3441) following the manufacturer's instructions.

## Extracellular field recordings

Hippocampal slices were prepared from 3-month-old mice (tamoxifen-injected at 8 weeks). Slices of mice were obtained from four different genotypes; $Slc9a6^{fl}$;$CAG$-$Cre^{ERT2}$ mice or $Slc9a6^{fl}$ mice with $Apoe^{APOE3}$ or $Apoe^{APOE4}$. The brains were quickly removed and placed in ice-cold high sucrose cutting solution (in mM: 110 sucrose, 60 NaCl, 3 KCl, 1.25 NaH$_2$PO$_4$, 28 NaHCO$_3$, 0.5 CaCl$_2$, 5 glucose, 0.6 ascorbic acid, 7 MgSO$_4$), bubbled with a gas mixture of 95 % O$_2$ and 5 % CO$_2$ for oxygenation. 350 µm transverse sections were cut using a vibratome (Leica). Slices were transferred into an incubation chamber containing 50 % artificial cerebrospinal fluid (aCSF, in mM: 124 NaCl, 3 KCl, 1.25 NaH$_2$PO$_4$, 26 NaHCO$_3$, 10 D-glucose, 2 CaCl$_2$, 1 MgSO$_4$) and 50 % sucrose cutting solution, oxygenated with 95% O$_2$/5% CO$_2$. Slices were transferred into an oxygenated interface chamber and perfused with aCSF with or without Reelin (2 µg/ml). The stimulating electrode was placed on the Schaffer-collateral of the CA1-pyramidal neurons and the recording electrode on the dendrites of the CA3-pyramidal neurons. Once baseline was stably recorded for 20 min, theta burst was applied, and traces collected for an hour. For stimulation concentric bipolar electrodes (FHC, catalog no CBBRC75) were placed into the stratum radiatum. Stimulus intensity was set at 40–60% of maximum response and delivered at 33 mHz through an Isolated Pulse Stimulator (A-M Systems, Model 2100). A custom written program in LabView 7.0 was used for recording and analysis of LTP experiments. A theta burst (TBS; train of four pulses at 100 Hz repeated 10 times with 200 ms intervals; repeated five times at 10 s intervals) was used as a conditioning stimulus.

## Statistical methods

Data were expressed as the mean ± SEM and evaluated using two-tailed Student's t-test for two groups with one variable tested and equal variances, one-way analysis of variance (ANOVA) with Dunnett's post hoc or Tukey's post hoc for multiple groups with only variable tested, or two-way ANOVA with Sidak's post hoc for plaque quantification (two independent variables of NHE6 genotype and plaque classification). The differences were considered to be significant at p < 0.05 (*p < 0.05, **p < 0.01, ***p < 0.001). Software used for data analysis was ImageJ (NIH), LabView7.0 (National Instruments), Odyssey Imaging Systems (Li-Cor), Prism7.0 (GraphPad Software).

## Acknowledgements

This work was supported by NIH grants R37 HL063762, R01 NS093382, R01 NS108115, and RF1 AG053391 to JH and 1F31 AG067708-01 to CHW as well as funding from the Darrell K Royal Research Fund to MD. While this work was ongoing, JH was further supported by the Bright Focus Foundation (A20135245) and (A2016396S); Harrington Discovery Institute; and Circle of Friends Pilot Synergy Grant; and the Blue Field Project to Cure FTD. We are indebted Rebekah Hewitt, Barsha Subbha, Huichuan Reyna, Issac Rocha, Tamara Terrones, Emily Boyle, Alisa Gilloon, Travis Wolff, and Eric Hall for their excellent technical assistance. We thank Dr Yuan Yang for creating the NHE6-FLAG plasmid

and the UTSW Whole Brain Microscopy Facility (WBMF) in the Department of Neurology and Neuro-therapeutics for assistance with slide scanning. The WBMF is supported by the Texas Institute for Brain Injury and Repair (TIBIR). John Shelton and the UT Southwestern's Histopathology Core provided help with paraffin sectioning as well as H&E and Thioflavin S staining. We thank Wolfgang Scholz for providing EMD87580.

## Additional information

### Competing interests

Xunde Xian, Joachim Herz: Inventor of Patent: https://patents.google.com/patent/US20110136832A1/en. The other authors declare that no competing interests exist.

### Funding

| Funder | Grant reference number | Author |
| --- | --- | --- |
| National Institutes of Health | HL063762 | Joachim Herz |
| BrightFocus Foundation | A20135245 | Joachim Herz |
| National Institute on Aging | 1F31AG067708-01 | Connie H Wong |
| Darrell K Royal Research Fund | | Murat S Durakoglugil |
| Harrington Discovery Institute | | Joachim Herz |
| Circle of Friends Pilot Synergy | | Joachim Herz |
| Blue Field Project to Cure FTD | | Joachim Herz |
| National Institutes of Health | NS093382 | Joachim Herz |
| National Institutes of Health | NS108115 | Joachim Herz |
| National Institutes of Health | AG053391 | Joachim Herz |
| BrightFocus Foundation | A2016396S | Joachim Herz |

The funders had no role in study design, data collection and interpretation, or the decision to submit the work for publication.

### Author contributions

Theresa Pohlkamp, Conceptualization, Formal analysis, Funding acquisition, Investigation, Methodology, Resources, Supervision, Validation, Visualization, Writing – original draft, Writing – review and editing; Xunde Xian, Conceptualization, Investigation, Methodology, Resources, Validation, Visualization; Connie H Wong, Conceptualization, Formal analysis, Funding acquisition, Investigation, Methodology, Resources, Validation, Visualization, Writing – original draft, Writing – review and editing; Murat S Durakoglugil, Formal analysis, Investigation, Methodology, Resources, Validation, Visualization; Gordon Chandler Werthmann, Formal analysis, Investigation, Methodology, Validation; Takaomi C Saido, Charles L White, Resources; Bret M Evers, Jade Connor, Investigation; Robert E Hammer, Methodology; Joachim Herz, Conceptualization, Funding acquisition, Investigation, Project administration, Resources, Supervision, Validation, Writing – original draft, Writing – review and editing

### Author ORCIDs

Theresa Pohlkamp http://orcid.org/0000-0003-3923-1917
Connie H Wong http://orcid.org/0000-0002-6452-7966
Murat S Durakoglugil http://orcid.org/0000-0003-4483-8166

Bret M Evers  http://orcid.org/0000-0001-5686-0315
Charles L White III,  http://orcid.org/0000-0002-3870-2804
Joachim Herz  http://orcid.org/0000-0002-8506-3400

## Ethics

All animal procedures were performed according to the approved guidelines (Animal Welfare Assurance Number D16-00296) for Institutional Animal Care and Use Committee (IACUC) at the University of Texas Southwestern Medical Center at Dallas.

## Decision letter and Author response

Decision letter https://doi.org/10.7554/eLife.72034.sa1
Author response https://doi.org/10.7554/eLife.72034.sa2

---

# Additional files

## Supplementary files

• Transparent reporting form

## Data availability

All relevant data are included in the manuscript.

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
