## [Decision Letter]

**Acceptance summary:**

This paper is of interest to a broad range of neuroscientists, particularly those interested in ApoE biology and Alzheimer's disease (AD), as it reveals a novel mechanism that counteracts AD-linked amyloid plaque burden and synapse dysfunction in mice. Overall, the methodology is sound, sophisticated, and employs animal models that more closely mimic human diseases, and the results are compelling.

**Decision letter after peer review:**

[Editors’ note: the authors submitted for reconsideration following the decision after peer review. What follows is the decision letter after the first round of review.]

Thank you for submitting your work entitled "Endosomal Acidification by NHE6-depletion Corrects ApoE4-mediated Synaptic Impairments and Reduces Amyloid Plaque Load" for consideration by *eLife*. Your article has been reviewed by 3 peer reviewers, one of whom is a member of our Board of Reviewing Editors, and the evaluation has been overseen by a Reviewing Editor and a Senior Editor. The following individual involved in review of your submission has agreed to reveal their identity: Eric Morrow (Reviewer #2).

We are sorry to say that, after consultation with the reviewers, we have decided that your work will not be considered further at this time for publication by *eLife*.

We would like to emphasize that all 3 reviewers found considerable strengths in your work, particularly the marked reductions in plaques in mice in which NHE6 was ablated or reduced. However, the lack of evidence that microglia and/or astrocytes are activated, take up Abeta, and mediate the plaque clearance in your studies leaves room for alternative explanations that were not addressed. Therefore, the data as they stand now do not strongly confirm the proposed mechanistic hypothesis put forward in your manuscript. This and other concerns listed below in the Reviewers' comments may take substantial time to satisfactorily address, which was the basis for our decision. If you decide to perform additional studies that address these concerns, we encourage you to consider submitting your thoroughly revised manuscript again to *eLife*.

*Reviewer #1:*

In this study, the authors confirm and extend their previous work demonstrating that ApoE4, a major risk for Alzheimer's disease, impairs endocytic recycling of membrane receptors, leading to synaptic dysfunction. Previously, the authors demonstrated in vitro that upon binding to ApoER2 at the plasma membrane and internalization, ApoE4 along with ApoER2 and glutamate receptors become trapped in the early endosome due to the similarity between the isoelectric point of ApoE4 and the pH of early endosomes. Enhancing acidification by inhibiting NHE6, a proton leak channel in the early endosome, restored vesicle recycling and improved synaptic plasticity in AD extract-treated hippocampal slices from ApoE4-KI mice. In the current study, the authors create and use novel NHE6 germline knockout and conditional knockout mouse lines to reduce NHE6 expression and enhance acidification of early endosomes. They confirm their previous findings and also extend their work by crossing NHE6 KO or cKO mice to a knockin, humanized APP mouse that expresses mutant human amyloid precursor protein under control of the endogenous APP promoter, alone or crossed with ApoE4-KI mice. In both cases, reduction of NHE6 resulted in increased Iba1-expressing microglia and GFAP-expressing astrocytes as well as a reduction in plaques. Together these findings highlight the importance of ApoE4's detrimental effect on endosomal recycling in vivo, with consequences for accumulation of AD-related pathology. While the studies presented are well-done and robust, some mechanistic links are missing, which make it difficult to fully support the conclusions drawn.

Major strengths of this paper include:

1. The creation of novel germline KO and conditional KO NHE6 mice allow for a number of in vivo investigations that would be difficult to complete otherwise. These mice represent a valuable resource to the field.

2. Use of the NHE6-KO and cKO mice to confirm the previous findings (that used pharmacological inhibition of NHE6) that enhancing endosomal acidification ameliorates ApoE4-induced deficits in vesicle cycling and synaptic plasticity in vivo. In addition, the finding that NHE6 ablation in APP and APP/ApoE4KI mice robustly reduced plaque accumulation is striking.

3. The demonstration that BACE-mediated production of APP CTFs is unaltered by NHE6 ablation supports the conclusion that the reduction in plaques is unlikely due to reduction in Abeta generation, but more likely due to clearance of Abeta.

Weaknesses of this paper include:

1. The authors conclude that reducing NHE6 clears plaques by activating resident microglia, shifting them from a dormant state to a damage-associated activated state that phagocytoses Abeta plaques. However, there is no data presented to demonstrate this. In a supplemental figure, the authors show there are more Iba1-expressing microglia and GFAP-expressing astroctyes in APP mice and in APP/ApoE4KI mice in which NHE6 has been ablated, but this does not prove that this is the mechanism by which plaques are cleared.

2. The mechanisms underlying the increase in Iba1 and GFAP are not clear. The authors cite a previous paper from another group that demonstrated in their own NHE6 KO mice, there was an increase in GFAP and in activated microglia expressing CD68, which may relate to the cell loss in hippocampus and other brain regions documented in those mice. However, in the current study, the authors indicate that in their NHE6 KO lines, there is no overt cell loss. It is therefore unclear how reductions in NHE6 expression lead to microglial/astrocyte activation. This is an important point to work out, since the authors conclude that it is microglial activation that is responsible for the reduction in Abeta plaques.

3. What might be some of the underlying explanations be for the differences between the published NHE6-KO mice, which has fairly widespread cell loss, and the current KO mice generated in this paper, which did not exhibit noticeable cell loss in brain regions other than the cerebellum?

4. There are a number of mechanistic links that have not been worked out, as indicated above. Until these links are identified and characterized, a number of the conclusions drawn by the authors are not yet supported.

Specific suggestions for authors:

1.To better assess whether the Iba1-expressing microglia are truly activated, CD68 should be stained. It would also be extremely compelling to stain for Abeta and demonstrate increased Abeta inside of microglia.

2. In order to conclude that NHE6 ablation clears plaques BY activating microglia, the authors should deplete microglia and then see whether there is still an effect on plaque load. If there is, that would firmly support their hypothesis; but if eliminating microglia has no effect on plaques, it would suggest that there are other mechanisms at play.

3. Careful assessment of cell viability/death in the NHE6 mice (and related crosses) should be done. It is puzzling why the NHE6 KO lines here would show such differences in level of cell loss, relative to the study by Xu et al. It is an important point though, because (1) it could help provide a mechanism by which microglia get activated. (2) it is necessary to fully appreciate the LTP studies – the Xu et al. paper indicated cell loss in the hippocampus, including the CA3-CA1 synapses, but there was no cell loss in KO or cKO described in this paper.

4. Some speculation is warranted to discuss the possible mechanisms that lead from loss of NHE6 to activation of microglia, since it is what the authors conclude is happening.

*Reviewer #2:*

This is a strong and interesting manuscript which examines innovative new hypotheses that have broad relevance to Alzheimer's disease pathophysiology as well as potential new therapeutics. Pohlkamp, Herz and colleagues study the role NHE6 in several orthogonal studies related to production and deposition of amyloid plaques. The use of various different experimental approaches as well as the use of advance mouse genetics is a strength.

The authors demonstrate several important findings that are robustly supported by the data including: late loss of NHE6 leads to Purkinje cell degeneration; recycling defects in surface receptors relevant to AD and APOE4, namely APOER2 and Glu receptors is improved by deletion of NHE6; NHE6 KO restores Reelin enhancement of LTP inhibited by APOE4; and profound decrease in plaque deposition due to NHE6 mutation.

The data are well presented in general and compelling. There are many strengths. The PC findings are important in the field of Christianson Syndrome. The reductions in plaque load in the NHE6 null brains are VERY interesting. The mouse genetics, including the conditional mutation -- presentation of a new cKO NHE6 mouse, the humanized Abeta and APOE4 alleles, are truly elegant. Some of the experiments are uniquely supported by the prior findings from the lab relating to Reelin effects on endocytosis and trafficking and effects on LTP, and this is a very important strength. I do not see major weaknesses with the experiments as presented.

I believe that this work will have broad interest and this work and prior work of the Herz lab in the area of NHE6 as it may relate to therapeutics in AD is developing into a unique niche with potential strong impact in AD therapeutics.

*Reviewer #3:*

In this manuscript, Pohlkamp, Xian, Wang et al. investigated the role of the sodium-hydrogen exchanger NHE6 in synaptic plasticity and Aβ plaque load in a mouse model of Alzheimer's disease (AD) in the presence or absence of Apolipoprotein E4 (APOE4), a major genetic risk factor for sporadic AD. They initially report that NHE6 deletion causes cerebellar neurodegeneration. They find that genetic deletion of NHE6 alleviates impairments in reelin-induced synaptic plasticity in mice expressing human APOE4. The main novelty of this study is that NHE6 suppression significantly reduced amyloid plaque load in a mouse model of AD expressing humanized Aβ, either in the presence or absence of ApoE4. This is interesting, as it potentially opens new roads to understand and control amyloid pathology in the AD brain. Although the data are intriguing and relevant for the community, some issues need to be addressed so that conclusions are justified by data:

1) The leading hypothesis of this work is that APOE4 impairs synapse function through prolonged association with endosomes, thereby making brain cells vulnerable to AD-related pathological changes. However, the positive effects of NHE6 in a mouse model of Aβ accumulation occurs regardless of APOE4. This suggests that NHE6 may contribute to pathology by mechanisms other than APOE4-mediated retention of endosomal trafficking.

2) With the current data, it is not possible to exclude possible nonspecific effects resulting from NHE6 genetic deletion. Additional experiments to measure the endosomal pH would add support to the hypothesis.

3) The authors attribute reduced amyloid plaque load in NHE6-deficient APP KI mice to increased glial responses, which would promote plaque clearance. This is a very interesting hypothesis, but it is not supported by the experimental data reported in Supplemental Figure 6. Additional experimentation is needed to more thoroughly characterize astrocytic and microglial phenotypes caused by NHE6 genetic depletion in APP KI mice. Functional assays, including cytokine release, nitric oxide production (Griess reaction), and Aβ uptake experiments would be desired to strengthen these conclusions.

4) The authors demonstrate that global or conditional NHE6 deletion causes severe Purkinje cell loss in the mouse cerebellum (Figure 2). Although the authors included representative images of H&E staining indicating no gross histological abnormalities (Supplemental Figure 3), a more detailed investigation is required to assess neuronal survival in the hippocampus and cortex upon NHE6 suppression, given the relevance of these regions to AD pathology. Indeed, previous evidence (Xu et al., eNeuro, 2017) showed that NHE6 depletion leads to significant cortical and hippocampal atrophy, in addition to the cerebellum. Could the reductions in plaque load in NHE6 depleted mice (Figure 5, 6; Supplemental Figure 5) be somehow a reflection of neuronal loss? It is important that the authors discuss this issue in the manuscript.

5) Even though it is an important control assessment, data on cerebellar neurodegeneration (in addition to eventual new data on other brain regions to be included in the manuscript) could be moved to the supplementary file. Conversely, data on glial activation (in addition to eventual new data) could be moved to the main figures.

6) The authors should thoroughly revise their manuscript to make it more concise and straightforward. In particular, the Introduction and Discussion are excessively long and include several pieces of information that are not essential to understand the study.

7) It would be helpful to include a paragraph discussing the limitations of this study in the Discussion. In a revised Discussion section, it would also be relevant to comment on previous studies assessing potential behavioral, neuroinflammatory, and neurodegenerative phenotypes caused by NHE6 disruption (e.g. Xu et al., eNeuro, 2017; Petitjean et al., Pain, 2020) in mice. Authors should also comment on the apparently contradictory findings by Prasad and Rao, PNAS, 2018.

8) Statistical analyses need to be particularly revised for consistency. Data normality should be assessed, and suitable tests should be performed for each data set. Sample size description is often conflicting between graphs and figure legends, and the appropriate statistical tests have not been performed in all cases. For the sake of transparency, all bar graphs should be replaced by scatter dot plots.

9) A remaining question is whether NHE6 deletion does provide any cognitive benefit to APP KI mice.

[Editors’ note: further revisions were suggested prior to acceptance, as described below.]

Thank you for resubmitting your work entitled "NHE6-Depletion Corrects ApoE4-Mediated Synaptic Impairments and Reduces Amyloid Plaque Load" for further consideration by *eLife*. Your revised article has been evaluated by Jonathan Cooper (Senior Editor) and three reviewers, one of whom is a member of our Board of Reviewing Editors.

The manuscript has been improved by the inclusion of new experiments and toning down of the text and conclusions drawn to better reflect what the data demonstrate. However, there are some remaining issues that need to be addressed, without additional experiments, as outlined below.

1. The new experiments that co-stained Abeta and Iba1 indicate that loss of NHE6 does not increase microglia around plaques, nor does it increase Abeta inside of microglia. The authors state that the results suggest increased efficiency of microglia in degrading Abeta. However, those results are also consistent with a lack of a role of microglia in the reduction of plaques in this case. Results from other studies have demonstrated that microglia can clear plaques, so this might be the most logical conclusion here as well, but no data to support this conclusion is presented in this manuscript. Therefore, the following is suggested:

1a. Further toning down of the manuscript; for example, at the end of the last Results section (page 20). The authors should remove the last statement that the mechanisms that lead to reduced plaque load in NHE6 deficient APP mice may involve increased catabolic rate, brought about by accelerated acidification and vesicular trafficking of early endosomes. There are two statements there for which the authors have not presented evidence – (1) that there is an increased catabolic rate, and (2) that accelerated acidification and endosomal trafficking increased catabolic rate. If the authors wish, they can discuss these possibilities in the discussion, and clearly state that these are speculations, but these statements should not be in the Results section.

1b. Clearer summary and discussion of the relative effects of NHE6 in neurons vs microglia or astrocytes.

1c. In the first paragraph of the discussion, the authors state that loss of NHE6 "suppressed" amyloid deposition. However, the data do not clearly distinguish whether there is reduction of deposition or enhancement of clearance. Perhaps saying that "loss of NHE6 reduces Abeta accumulation" may be more appropriate.

2. The Results sections describing the new data for (lack of) neuronal loss and co-staining of Abeta and Iba1/GFAP need to be edited and streamlined. There seems to be some inconsistency in the references to figures. Either the references are not in the right order, or the way they are referred to is not as streamlined as it could be.

3. The new discussion of the contradictory results related to Prasad and Rao 2018 highlights the difference between tissue culture versus genetically engineered mice as a possible reason underlying the discrepant results, but how about the difference of astrocyte vs neurons? Prasad an Rao overexpressed NHE6 in astrocytes, whereas the current manuscript manipulated primary neurons or mice. Such discussion would also enhance the readers' understanding of the relative roles of NHE6 in neurons vs microglia and astrocytes.

4. The authors state: "Xu et al. (2017) reported neuronal loss in the cortex and hippocampus of NHE6-KO mice, which we were able to reproduce in our germline NHE6-KO model (Supplemental Figure S3)."

Xu et al. (2017) examined tissue area. Xu et al. (2017) did not examine "neuronal loss" in cortex and hippocampus. The authors might agree that neuronal loss would reflect an observation wherein there is a measurable loss of neurons, ie counts of neurons are reduced. Instead, the studies reflect measurements of tissue area.

This underscores another new and important finding in the manuscript as stated: "We measured brain size, cortical thickness, hippocampal area, and CA1 thickness. In contrast to germline NHE6-KO mice (Supplemental Figure S3) none of the analyzed parameters differed significantly between NHE6- cKO and controls (Figure 5 E-I)."

These results could argue that the decrease tissue area in NHE6-KO, that is not seen in the NHE6-cKO, is due to undergrowth of cortex and hippocampus, ie the reduced tissue area in the NHE6-KO is a developmental effect due to loss of NHE6 during development. This would be predicted by the postnatal microcephaly seen in patients. This was also predicted by modeling in the Xu et al. paper: "The cerebrum data do fit an undergrowth-only model, with a similar degenerative rate as wild-type animals. The cerebellum strongly supports a mixed scenario of both undergrowth and enhanced neurodegeneration…" Further, Xu et al. describe that ventricle size is not vastly enlarged which might be predicted by neurodegeneration.

If the authors agree, the authors should consider editing their writing to reflect that Xu et al. did not look at "neuronal loss", and further, they may consider adding this interpretation of the difference between the tissue area in the NHE6-KO vs their NHE6-cKO to the discussion, ie that this may be a developmental effect.

5. Introduction: The authors may wish to consider shortening the first paragraph of the introduction, which is quite long (lines 2-43).

6. Abstract (line 13): change "amyloid" to "plaque load".

7. Experimental procedures:

7a. page 7, line 28: change "for primer" to "forward primer".

7b. page 8, lines 12-13: details (i.e. host, dilution, and catalog number) of antibodies raised against GFAP, IBA1, and Aβ should be provided.

7c. More details on the pH measurements should be provided. How many fields on each coverslip were quantified? How many vesicles per field were quantified? What was considered an experimental "n" (Supplemental Figure S1E-F)?

8. Results: In page 19 (lines 4-5) the conclusion is not related to the results presented in this paragraph. Results related to glial activation are only presented in the following paragraph.

9. Figure legends:

9a. Figure 2B: please specify the sample size.

9b. Figure 4: the phrase "Conditional knockout of NHE6 in ApoE4-KI mice attenuates reelin-enhanced long-term potentiation" is confusing and should be clarified.

9c. Supplemental Figure S1-E-F: statistical tests should be specified.

9d. Supplemental Figure S3: the reported sample sizes differ between the figure and the legend, which needs to be reconciled for accuracy.

10. Discussion: In page 20, lines 19-20: the phrase "hyper acidification of early endosomes occludes the effect of all ApoE forms on amyloid plaque formation" would be more accurate and easier to read if written as "hyper acidification of early endosomes prevents amyloid plaque formation independently of ApoE4".

---

## [Author Response]

[Editors’ note: the authors resubmitted a revised version of the paper for consideration. What follows is the authors’ response to the first round of review.]

Reviewer #1:[…]Weaknesses of this paper include:1. The authors conclude that reducing NHE6 clears plaques by activating resident microglia, shifting them from a dormant state to a damage-associated activated state that phagocytoses Abeta plaques. However, there is no data presented to demonstrate this. In a supplemental figure, the authors show there are more Iba1-expressing microglia and GFAP-expressing astroctyes in APP mice and in APP/ApoE4KI mice in which NHE6 has been ablated, but this does not prove that this is the mechanism by which plaques are cleared.

We apologize for the overstatement. We agree, we have not evaluated whether NHE6 depletion causes a signature of damage-associated microglia. Thus, we have removed this comment from the manuscript (abstract).

2. The mechanisms underlying the increase in Iba1 and GFAP are not clear. The authors cite a previous paper from another group that demonstrated in their own NHE6 KO mice, there was an increase in GFAP and in activated microglia expressing CD68, which may relate to the cell loss in hippocampus and other brain regions documented in those mice. However, in the current study, the authors indicate that in their NHE6 KO lines, there is no overt cell loss. It is therefore unclear how reductions in NHE6 expression lead to microglial/astrocyte activation. This is an important point to work out, since the authors conclude that it is microglial activation that is responsible for the reduction in Abeta plaques.

We agree that identifying the mechanism how NHE6 depletion causes glial activation is crucial. We and others show that germline NHE6 depletion causes glial activation. Moreover, our current data suggest that genetic deletion of NHE6 in both germline and from adulthood on causes glial activation. Neuronal cell loss is a potential explanation for glial activation. We found cerebellar Purkinje cell loss in both of our NHE6 mutant lines. As stated in the old version of our manuscript we find “Normal Gross Anatomical Brain Structure in Both NHE6-KO and NHE6cKO Mice” (Supplementary Figure S3). To address whether neuronal loss occurs in the hippocampus or cortex, as described for NHE6-KO mice in Xu et al., 2017, we measured neuronal loss in both NHE6 mutant lines. Comparable to Xu et al., in the NHE6-KO line we detect a reduction in total brain area, HC area, cortical thickness and CA1 thickness. By contrast, in our NHE6-cKO;APP-KI;ApoE4-KI mice we do not observe any neuronal loss when compared to NHE6-floxed,APP^NL-F^,ApoE4-KI littermate controls; however, we detect similar glial activation in the NHE6-cKO;APP^NL-F^;ApoE4-KI mice as compared to the germline NHE6KO,APP^NL-F^ mice. This suggests that the neuronal loss in the germline NHE6-KO model does not mediate glial activation. Lastly, we have removed the statement that the microglial activation is the reason why we detect Aβ reduction and included a discussion of our new findings.

3. What might be some of the underlying explanations be for the differences between the published NHE6-KO mice, which has fairly widespread cell loss, and the current KO mice generated in this paper, which did not exhibit noticeable cell loss in brain regions other than the cerebellum?

Our previous manuscript stated that there are no gross anatomical abnormalities in the NHE6KO mice. However, we appreciate the reviewer’s concerns as it prompted us to analyze neuronal loss in NHE6-KO versus NHE6-cKO mice. Besides Purkinje cell loss in both lines, and as stated above, we do detect cell loss in the hippocampus and cortex in our germline NHE6KO mouse model, but not in the tamoxifen induced NHE6-cKO mice.

4. There are a number of mechanistic links that have not been worked out, as indicated above. Until these links are identified and characterized, a number of the conclusions drawn by the authors are not yet supported.

We thank the reviewer for the constructive feedback. We have removed these conclusions.

Specific suggestions for authors:1.To better assess whether the Iba1-expressing microglia are truly activated, CD68 should be stained. It would also be extremely compelling to stain for Abeta and demonstrate increased Abeta inside of microglia.

We thank the reviewer for this suggestion. Whereas we had too much non-specific background with the CD68 staining (see Author response image 1), we successfully co-stained Aβ with Iba1 and GFAP similar to others (Feng, W., et al., Alzheimers Res. Ther., 2020; Monasor, L., et al., *ELife*, 2020; Pomilio, C., et al., Geroscience, 2020; Parhizkar, S., Nat. Neurosci., et al., 2019; Zhon, L., et al., Nat. Commun., 2019). We did not detect a significant difference between genotypes in Aβ inside microglia. In addition, we quantified the amount of microglia and astrocytes surrounding plaques, with no significant differences between genotypes (NHE6-KO;APP^NL-F^ and APP^NL-F^). As NHE6-KO mice show less plaque/Aβ-staining while having increased immunoreactivity for GFAP and Iba1, the fact that the proportion of microglia containing Aβ is equal to control indirectly suggests that the microglia are indeed more efficient in Aβ uptake and degradation. This hypothesis will be tested in future studies to evaluate Aβ uptake in primary microglia derived from NHE6-KO mice. We discussed our findings in the manuscript accordingly.

**Author response image 1. sa2fig1:** 

2. In order to conclude that NHE6 ablation clears plaques BY activating microglia, the authors should deplete microglia and then see whether there is still an effect on plaque load. If there is, that would firmly support their hypothesis; but if eliminating microglia has no effect on plaques, it would suggest that there are other mechanisms at play.

We thank the reviewer for this suggestion. We are currently in the process of obtaining the animal approval for depleting the microglia in our mouse model. Future experiments will allow us to evaluate whether depleting the activated microglial affects the reduction of Aβ plaques.

3. Careful assessment of cell viability/death in the NHE6 mice (and related crosses) should be done. It is puzzling why the NHE6 KO lines here would show such differences in level of cell loss, relative to the study by Xu et al. It is an important point though, because (1) it could help provide a mechanism by which microglia get activated. (2) it is necessary to fully appreciate the LTP studies – the Xu et al. paper indicated cell loss in the hippocampus, including the CA3-CA1 synapses, but there was no cell loss in KO or cKO described in this paper.

As stated above, we evaluated cell loss in both the NHE6-KO and NHE6-cKO lines. In contrast to germline NHE6-KO mice, we did not detect cell loss in the cortex or hippocampus of the tamoxifen induced NHE6-cKO line, which was used in the electrophysiological studies. We agree with the reviewer that these novel data are crucial to better understand if NHE6-loss is a primary or secondary factor to induce glia activation.

4. Some speculation is warranted to discuss the possible mechanisms that lead from loss of NHE6 to activation of microglia, since it is what the authors conclude is happening.

We have added more potential mechanism in the discussion.

Reviewer #3:In this manuscript, Pohlkamp, Xian, Wang et al. investigated the role of the sodium-hydrogen exchanger NHE6 in synaptic plasticity and Aβ plaque load in a mouse model of Alzheimer's disease (AD) in the presence or absence of Apolipoprotein E4 (APOE4), a major genetic risk factor for sporadic AD. They initially report that NHE6 deletion causes cerebellar neurodegeneration. They find that genetic deletion of NHE6 alleviates impairments in reelin-induced synaptic plasticity in mice expressing human APOE4. The main novelty of this study is that NHE6 suppression significantly reduced amyloid plaque load in a mouse model of AD expressing humanized Aβ, either in the presence or absence of ApoE4. This is interesting, as it potentially opens new roads to understand and control amyloid pathology in the AD brain. Although the data are intriguing and relevant for the community, some issues need to be addressed so that conclusions are justified by data:1) The leading hypothesis of this work is that APOE4 impairs synapse function through prolonged association with endosomes, thereby making brain cells vulnerable to AD-related pathological changes. However, the positive effects of NHE6 in a mouse model of Aβ accumulation occurs regardless of APOE4. This suggests that NHE6 may contribute to pathology by mechanisms other than APOE4-mediated retention of endosomal trafficking.

We agree with the reviewer that NHE6 depletion plays a protective role in AD both by protecting against synaptic impairments in ApoE4-KI mice and Aβ toxicity in an Aβ overproducing mouse model. This may reflect a beneficial effect of endosomal compartment acidification through NHE6 depletion. Our current work and studies by others (Fagan, A.M., et al., Neurobio. of Dis. 2002) show that human Aβ-overproducing ApoE4-KI mice generate plaques at a much later age than mice with wildtype, mouse ApoE, but the mechanism is unknown. Since both of our mouse models, NHE6-KO and NHE6-cKO;ApoE4-KI show a comparable reduction in plaque load, this might be the result of a maximally accelerated early endosomal maturation and cargo transport in the absence of NHE6. We elaborated on this in topic in the discussion of our manuscript.

2) With the current data, it is not possible to exclude possible nonspecific effects resulting from NHE6 genetic deletion. Additional experiments to measure the endosomal pH would add support to the hypothesis.

We agree with the reviewer’s concern and addressed this in the discussion accordingly.

3) The authors attribute reduced amyloid plaque load in NHE6-deficient APP KI mice to increased glial responses, which would promote plaque clearance. This is a very interesting hypothesis, but it is not supported by the experimental data reported in Supplemental Figure 6. Additional experimentation is needed to more thoroughly characterize astrocytic and microglial phenotypes caused by NHE6 genetic depletion in APP KI mice. Functional assays, including cytokine release, nitric oxide production (Griess reaction), and Aβ uptake experiments would be desired to strengthen these conclusions.

We thank the reviewer for this valuable feedback. In our revised manuscript, we evaluated whether there is a change of microglial Ab content in the NHE6 depletion mouse model. We also quantified the immunoreactivity of Iba1 and GFAP in plaque areas. We found no change between NHE6-KO or control littermate APP^NL-F^ controls when we co-stained with Aβ and Iba1 (microglia) or GFAP (astrocyte). However, when considering a massively reduced Aβ signal in NHE6-KO brains overall, yet the proportion of microglia containing Aβ is comparable to control, this indirectly indicates that NHE6 deficient microglia are more efficient in Aβ uptake and degradation. We agree with the reviewers that future studies will be required to evaluate Aβ uptake in primary microglia derived from NHE6-KO mice to properly conclude that the reduction of Aβ is mediated by enhance glial activation. Thus, we have adjusted our conclusions in the manuscript accordingly.

4) The authors demonstrate that global or conditional NHE6 deletion causes severe Purkinje cell loss in the mouse cerebellum (Figure 2). Although the authors included representative images of H&E staining indicating no gross histological abnormalities (Supplemental Figure 3), a more detailed investigation is required to assess neuronal survival in the hippocampus and cortex upon NHE6 suppression, given the relevance of these regions to AD pathology. Indeed, previous evidence (Xu et al., eNeuro, 2017) showed that NHE6 depletion leads to significant cortical and hippocampal atrophy, in addition to the cerebellum. Could the reductions in plaque load in NHE6 depleted mice (Figure 5, 6; Supplemental Figure 5) be somehow a reflection of neuronal loss? It is important that the authors discuss this issue in the manuscript.

We thank the reviewer for this suggestion. We have now measured brain area, hippocampal area, cortical and CA1 thickness. Comparable to Xu et al., we detect a reduction in total brain area, HC area, cortical thickness and CA1 thickness. Contrary, in our NHE6-cKO;APP^NFL^;ApoE4-KI mice we do not see any neuronal loss; however, we detect similar plaque reduction and glial activation in the NHE6-KO;APP^NL-F^ mice. These findings suggest that the neuronal loss does not mediate the reduction in plaque load or glial activation. We discussed our findings in our manuscript accordingly.

5) Even though it is an important control assessment, data on cerebellar neurodegeneration (in addition to eventual new data on other brain regions to be included in the manuscript) could be moved to the supplementary file. Conversely, data on glial activation (in addition to eventual new data) could be moved to the main figures.

We thank the reviewer for the suggestion. We included new data on neuronal loss in additional brain regions of the NHE6 mutant mice. As specific findings on germline NHE6-KO mice are already published by others (and now reproduced by us), we decided to keep our results on NHE6-KO in the supplements, while keeping the novel results on NHE6-cKO in the main figures. Our novel data on glia and Aβ co-labeling are now included in the main manuscript, Figure 7.

6) The authors should thoroughly revise their manuscript to make it more concise and straightforward. In particular, the Introduction and Discussion are excessively long and include several pieces of information that are not essential to understand the study.

We thank the reviewer for this comment and shortened the introduction. While adding several points to our discussion, based on reviewers’ suggestions and novel findings, we also shortened it by removing the paragraph addressing NHE6 C-terminal interactions, since it is not relevant to understand the study.

7) It would be helpful to include a paragraph discussing the limitations of this study in the Discussion. In a revised Discussion section, it would also be relevant to comment on previous studies assessing potential behavioral, neuroinflammatory, and neurodegenerative phenotypes caused by NHE6 disruption (e.g. Xu et al., eNeuro, 2017; Petitjean et al., Pain, 2020) in mice. Authors should also comment on the apparently contradictory findings by Prasad and Rao, PNAS, 2018.

We thank the reviewer for this suggestion and have added new paragraphs to our discussion that also address limitations of our study. We also discussed the findings by Prasad and Rao (Prasad, H. and Rao, R., PNAS 2018).

8) Statistical analyses need to be particularly revised for consistency. Data normality should be assessed, and suitable tests should be performed for each data set. Sample size description is often conflicting between graphs and figure legends, and the appropriate statistical tests have not been performed in all cases. For the sake of transparency, all bar graphs should be replaced by scatter dot plots.

We thank the reviewer for pointing this out. We carefully went through the statistics and the numbers. Corrections have been made for figure legends 6, S5, and S6. Scatter dot plots have been provided for Figure 3 and 7.

9) A remaining question is whether NHE6 deletion does provide any cognitive benefit to APP KI mice.

We agree with the reviewer that behavioral insights would be compelling. However, APP^NL-F^ mice do not show behavioral abnormalities until at least 18 months of age (Saito et al., 2014). In addition, when we bred APP^NL-F^ with ApoE4 and ApoE3 mice, we were not able to detect differences in the Morris Water Maze test between ApoE genotype, either. Thus, we are unable to determine whether NHE6 depletion provides any cognitive benefits in the APP^NL-F^ mouse line. We addressed this in our revised discussion.

[Editors’ note: what follows is the authors’ response to the second round of review.]

The manuscript has been improved by the inclusion of new experiments and toning down of the text and conclusions drawn to better reflect what the data demonstrate. However, there are some remaining issues that need to be addressed, without additional experiments, as outlined below.1. The new experiments that co-stained Abeta and Iba1 indicate that loss of NHE6 does not increase microglia around plaques, nor does it increase Abeta inside of microglia. The authors state that the results suggest increased efficiency of microglia in degrading Abeta. However, those results are also consistent with a lack of a role of microglia in the reduction of plaques in this case. Results from other studies have demonstrated that microglia can clear plaques, so this might be the most logical conclusion here as well, but no data to support this conclusion is presented in this manuscript. Therefore, the following is suggested:1a. Further toning down of the manuscript; for example, at the end of the last Results section (page 20). The authors should remove the last statement that the mechanisms that lead to reduced plaque load in NHE6 deficient APP mice may involve increased catabolic rate, brought about by accelerated acidification and vesicular trafficking of early endosomes. There are two statements there for which the authors have not presented evidence – (1) that there is an increased catabolic rate, and (2) that accelerated acidification and endosomal trafficking increased catabolic rate. If the authors wish, they can discuss these possibilities in the discussion, and clearly state that these are speculations, but these statements should not be in the Results section.

We agree that this last statement should be removed from our Results section and added to the discussion (Page 24).

1b. Clearer summary and discussion of the relative effects of NHE6 in neurons vs microglia or astrocytes.

We have added a detailed discussion (Page 25) on the effects of NHE6 in various cell types (neurons, microglia or astrocytes).

1c. In the first paragraph of the discussion, the authors state that loss of NHE6 "suppressed" amyloid deposition. However, the data do not clearly distinguish whether there is reduction of deposition or enhancement of clearance. Perhaps saying that "loss of NHE6 reduces Abeta accumulation" may be more appropriate.

We have adjusted this statement in our discussion (Page 20).

2. The Results sections describing the new data for (lack of) neuronal loss and co-staining of Abeta and Iba1/GFAP need to be edited and streamlined. There seems to be some inconsistency in the references to figures. Either the references are not in the right order, or the way they are referred to is not as streamlined as it could be.

We apologize for any confusion and have made the requested modifications to help streamline the figures and text, which is located on page 18 and 19.

3. The new discussion of the contradictory results related to Prasad and Rao 2018 highlights the difference between tissue culture versus genetically engineered mice as a possible reason underlying the discrepant results, but how about the difference of astrocyte vs neurons? Prasad an Rao overexpressed NHE6 in astrocytes, whereas the current manuscript manipulated primary neurons or mice. Such discussion would also enhance the readers' understanding of the relative roles of NHE6 in neurons vs microglia and astrocytes.

We apologize for any confusion, and we made corrections to this paragraph. Previously, we referred to astrocytes when in fact Prasad and Rao used HEK293 cells (Prasad and Rao, 2015) to look at Ab generation. We have added additional text to explain the discrepant results (page 23) and included section for the readers about the relative roles of NHE6 in neurons versus microglia and astrocytes (Page 25, see Reviewer Comment 1b).

4. The authors state: "Xu et al. (2017) reported neuronal loss in the cortex and hippocampus of NHE6-KO mice, which we were able to reproduce in our germline NHE6-KO model (Supplemental Figure S3)."Xu et al. (2017) examined tissue area. Xu et al. (2017) did not examine "neuronal loss" in cortex and hippocampus. The authors might agree that neuronal loss would reflect an observation wherein there is a measurable loss of neurons, ie counts of neurons are reduced. Instead, the studies reflect measurements of tissue area.This underscores another new and important finding in the manuscript as stated: "We measured brain size, cortical thickness, hippocampal area, and CA1 thickness. In contrast to germline NHE6-KO mice (Supplemental Figure S3) none of the analyzed parameters differed significantly between NHE6- cKO and controls (Figure 5 E-I)."These results could argue that the decrease tissue area in NHE6-KO, that is not seen in the NHE6-cKO, is due to undergrowth of cortex and hippocampus, ie the reduced tissue area in the NHE6-KO is a developmental effect due to loss of NHE6 during development. This would be predicted by the postnatal microcephaly seen in patients. This was also predicted by modeling in the Xu et al. paper: "The cerebrum data do fit an undergrowth-only model, with a similar degenerative rate as wild-type animals. The cerebellum strongly supports a mixed scenario of both undergrowth and enhanced neurodegeneration…" Further, Xu et al. describe that ventricle size is not vastly enlarged which might be predicted by neurodegeneration.If the authors agree, the authors should consider editing their writing to reflect that Xu et al. did not look at "neuronal loss", and further, they may consider adding this interpretation of the difference between the tissue area in the NHE6-KO vs their NHE6-cKO to the discussion, ie that this may be a developmental effect.

We thank the reviewer for this valuable discussion of our findings and the important point clarifying Xu et al. We have revised discussion accordingly and included detailed discussion of the Xu et al. findings in the context of our results (Page 18, 19, and 25).

5. Introduction: The authors may wish to consider shortening the first paragraph of the introduction, which is quite long (lines 2-43).

We appreciate this comment and we have accordingly introduced line breaks into the first paragraph (Page 3).

6. Abstract (line 13): change "amyloid" to "plaque load".

We’ve changed this text in the abstract.

7. Experimental procedures:7a. page 7, line 28: change "for primer" to "forward primer".7b. page 8, lines 12-13: details (i.e. host, dilution, and catalog number) of antibodies raised against GFAP, IBA1, and Aβ should be provided.7c. More details on the pH measurements should be provided. How many fields on each coverslip were quantified? How many vesicles per field were quantified? What was considered an experimental "n" (Supplemental Figure S1E-F)?

We have added these experimental details to the experimental procedures.

8. Results: In page 19 (lines 4-5) the conclusion is not related to the results presented in this paragraph. Results related to glial activation are only presented in the following paragraph.

We have adjusted the text accordingly.

9. Figure legends:9a. Figure 2B: please specify the sample size.9b. Figure 4: the phrase "Conditional knockout of NHE6 in ApoE4-KI mice attenuates reelin-enhanced long-term potentiation" is confusing and should be clarified.9c. Supplemental Figure S1-E-F: statistical tests should be specified.9d. Supplemental Figure S3: the reported sample sizes differ between the figure and the legend, which needs to be reconciled for accuracy.

We apologize for the oversight and typos. We have adjusted the text accordingly.

10. Discussion: In page 20, lines 19-20: the phrase "hyper acidification of early endosomes occludes the effect of all ApoE forms on amyloid plaque formation" would be more accurate and easier to read if written as "hyper acidification of early endosomes prevents amyloid plaque formation independently of ApoE4".

We have changed this text.